**Subject Category:**
Biology (whole organism)

ecology/environmental science

homogenization, carbonate production, reef function, coral reefs, reef zones

**Author for correspondence:**
Lorenzo Alvarez-Filip
e-mail: lorenzo@cmarl.unam.mx

# Functional consequences of the long-term decline of reef-building corals in the Caribbean: evidence of across-reef functional convergence

Nuria Estrada-Saldívar[1,3], Eric Jordán-Dalhgren[2], Rosa E. Rodríguez-Martínez[2], Chris Perry[4] and Lorenzo Alvarez-Filip[1]

[1]Biodiversity and Reef Conservation Laboratory, Unidad Académica de Sistemas Arrecifales, Instituto de Ciencias del Mar y Limnología, and [2]Coral Ecology Laboratory, Unidad Académica de Sistemas Arrecifales, Instituto de Ciencias del Mar y Limnología, Universidad Nacional Autónoma de México, Puerto Morelos, Quintana Roo, México
[3]Posgrado en Ciencias del Mar y Limnología, Universidad Nacional Autónoma de México, Avenida Ciudad Universitaria 3000, CP 04510 Coyoacán, Ciudad de México, México
[4]Geography, College of Life and Environmental Sciences, University of Exeter, Exeter EX4 4RJ, UK

(iD) RER-M, 0000-0002-1383-1108; LA, 0000-0002-5726-7238

Functional integrity on coral reefs is strongly dependent upon coral cover and coral carbonate production rate being sufficient to maintain three-dimensional reef structures. Increasing environmental and anthropogenic pressures in recent decades have reduced the cover of key reef-building species, producing a shift towards the relative dominance of more stress-tolerant taxa and leading to a reduction in the physical functional integrity. Understanding how changes in coral community composition influence the potential of reefs to maintain their physical reef functioning is a priority for their conservation and management. Here, we evaluate how coral communities have changed in the northern sector of the Mexican Caribbean between 1985 and 2016, and the implications for the maintenance of physical reef functions in the back- and fore-reef zones. We used the cover of coral species to explore changes in four morpho-functional groups, coral community composition, coral community calcification, the reef functional index and the reef carbonate budget. Over a period of 31 years, ecological homogenization occurred

between the two reef zones mostly due to a reduction in the cover of framework-building branching (*Acropora* spp.) and foliose-digitiform (*Porites porites* and *Agaricia tenuifolia*) coral species in the back-reef, and a relative increase in non-framework species in the fore-reef (*Agaricia agaricites* and *Porites astreoides*). This resulted in a significant decrease in the physical functionality of the back-reef zone. At present, both reef zones have negative carbonate budgets, and thus limited capacity to sustain reef accretion, compromising the existing reef structure and its future capacity to provide habitat and environmental services.

## 1. Introduction

The three-dimensional structures, provided by reef-building corals, sustain one of the most biodiverse and socio-economically important ecosystems on the planet [1,2]. However, over the last 40 years, the average live coral cover on tropical reefs has declined significantly, with the Caribbean being among the regions that has experienced the most severe changes since the 1970s [3,4]. The causes of coral cover decline include a combination of local and global anthropogenic impacts including overfishing, coastal development and associated pollution and rising sea temperatures [5–7]. This decline has compromised the future capacity of coral reefs to sustain structural complexity (and with that the biota that depends on the structure), to maintain many ecosystem services and to keep up with sea-level rise [2,8–10]. These changes can occur either when vertical coral reef growth is halted or inhibited (i.e. reef 'turn off' occurs; [11,12]), when high rates of biological, chemical and physical processes drive net erosion of the underlying reef structure [13–15], or in response to direct impacts such as hurricanes through the breakage of coral skeletons [16]. The resultant loss of reef three-dimensional structures has serious implications for the local economy, such as fishing and tourism, and since wave attenuation functions are reduced [17], it can also result in changing coastal wave energy exposure [9,18–21].

In the western Atlantic, a few species of framework-building corals have dominated coral reef habitats throughout the region since at least the late Pleistocene [22–25]. Ecological and geological records of reef-building corals show that *Acropora* was historically one of the dominant coral genera and a major shallow-water reef-builder [4,24,26]. However, populations of acroporids declined considerably between the 1980s and 1990s due to the white-band disease [26–28], and since then, very little recovery has been reported [29,30]. After the acroporid dies off, massive corals in the genus *Orbicella* remained as major Caribbean reef builders; however, their populations have decreased in the last two decades in many areas mainly due to diseases and bleaching impacts [31–34]. What is most concerning is that remnant populations of reef-building corals are being affected by new emerging diseases [35–37] and thermal stress events [38,39].

The decline of the major reef-building coral species across the Caribbean has been accompanied by a relative increase in the abundance of non-framework coral species, such as *Agaricia agaricites* and *Porites astreoides* [40–43]. This group of new dominant species is characterized by small-sized colonies that do not contribute importantly to the reef framework [44,45]. The shift in the relative dominance patterns of Caribbean coral communities is strongly linked to the life-history strategies of corals and how they cope with rapidly changing environmental conditions [46,47]. By taking into account these attributes, alongside environmental variables, it should, therefore, be possible to hypothesize about future changes in coral assemblages and how this will affect reef functioning, especially the potential of corals to accrete three-dimensional structures and to provide habitat [48–50]. A serious consequence of a reduced abundance of important reef-building species will be reduced reef-carbonate budgets. Along with this decline, rates of bioerosion may become increasingly important controls on overall budgets states [44,51]. Indeed, if coral carbonate production rates are sufficiently suppressed, carbonate budget will transition into states of net erosion compromising future reef accretion potential and endangering current reef structural complexity [52].

Key to understanding changes in coral communities are long-term studies, but this remains a challenging issue, partly because community shifts may occur slowly and most measures of assessment rely upon the comparison of present status to a defined past reference condition [53]. Furthermore, robust datasets that can support historical timescale assessments are not numerous because past reference baselines are sparse or difficult to construct. Despite this, the use of historical records can be especially important for understanding contemporary ecological transitions and for aiding predictions of future changes [54,55]. In this study, we use data collected in 1985 and 2016 from back- and fore-reef sites in the northern Mexican Caribbean, in order to assess changes in the

community structure of reef-building corals and to determine the implications for reef functioning. We explore decadal-scale changes in community composition and track the trajectories of the main coral groups that influence the reef structure with respect to colony shape and function (framework-building branching, foliose-digitiform, massive and non-framework). We also evaluate changes in the physical functionality of reefs (defined as the capacity to sustain reef framework, a positive carbonate budget and the potential of reef accretion), by using species identity and composition to estimate changes in calcification rates and reef-carbonate budgets through time.

# 2. Material and methods

## 2.1. Study area

We conducted this study in the Puerto Morelos reef system, located in the northern part of the Mexican Caribbean. This is a fringing reef system that stretches parallel to the coast (between 1 and 3 km) in a semi-continuous formation. This reef system has an identifiable zonation with a back-reef, reef crest and fore-reef that is mostly strongly influenced by wave exposure and light penetration [56]. Historically, it had a well-developed back-reef and reef-crest that were dominated by *Acropora palmata*, which contributed greatly to the structural complexity of the reef; while the fore-reef was mostly of low relief (limited framework development), gentle sloping and colonized by sparse coral grounds, and grades gradually at a depth of approximately 20–25 m into an extensive sand platform [57,58]. The most conspicuous components of the fore-reef zone were octocorals, macroalgae and small coral heads [58]. This type of morphology favours sand accumulation and its resuspension during storms or hurricanes, which makes live scleractinian coral cover on the fore-reef zone sparse [59,60]. During the course of our study (1985–2016), coastal development in our study area has increased very rapidly as tourism became the main economic activity [61]. Coastal development poses several threats to the well-being of coastal systems including the increase in nutrients and pollutant levels in coastal waters due to the general lack of sewage treatment plants in the area (with the exception of some hotels), an underground water circulation system that outfalls in mangrove wetlands and submarine springs and the seepage through the sand bar in response to rain inputs [7,62]. Sedimentation is not a major problem due to the lack of superficial rivers in the Yucatan peninsula. In 1998, Puerto Morelos reef system was declared a marine protected area and, on average, it receives *ca* 200 000 visitors per year [63].

## 2.2. Data collection

This study compares data obtained in 1985 and in 2016. Data for 1985 were obtained by Jordan-Dahlgren [57], which conducted an assessment of the coral reefs along the Mexican Caribbean across the main reef-zones, down to approximately 20–25 m depth, and generated detailed maps of the coral-reef distribution along the Mexican Caribbean coast. The data obtained from this historical dataset for the northern part of the Mexican Caribbean were compared to recent estimates of reef condition. The methodology used to estimate coral cover at species level was slightly different in 1985 and 2016 but, despite these differences, the two methods employed are known to produce relatively similar estimates of benthic cover [64]. In 1985, surveys were conducted by means of line intercept transects [65] at three zones on each reef site: back-reef, reef-crest and fore-reef [57]. In the fore-reef, surveys were conducted at four depths (5, 10, 15 and 20 m). At each zone (back-reef and reef-crest) or depth in the case of the fore-reef, five, 20 m long transects were placed haphazardly, perpendicular to the coast, separated from each other by 5–25 m. The transects were delimited by plastic chains, with a 2.73 cm size chain link that followed the contour of the bottom. All scleractinian corals below the chain were measured using the chain link as the measurement unit. In 2016, surveys were conducted in the back-reef and part of the reef crest (between 2 and 5 m deep) and in the fore-reef (between 6 and 13 m deep) zones following the AGRRA protocol v. 5.5 [66]. At each zone, between six and eight, 10 m long transects were placed haphazardly parallel to the coast on each reef site and surveyed using the point intercept method to determine benthic cover, including hard coral cover identified to species level; transects were separated from each other by 5–25 m.

   In 1985, the geographical coordinates of each reef site were not obtained as the surveys were before the general use of GPS in scientific research, but to try to ensure realistic site comparisons, we only selected sites in 1985 and 2016 where geographical location could be accurately constrained based on the original maps that indicated surveying locations, depth and distinct geomorphological structures [57]. For the present study, we compared only data from the back-reef zone and the approximately

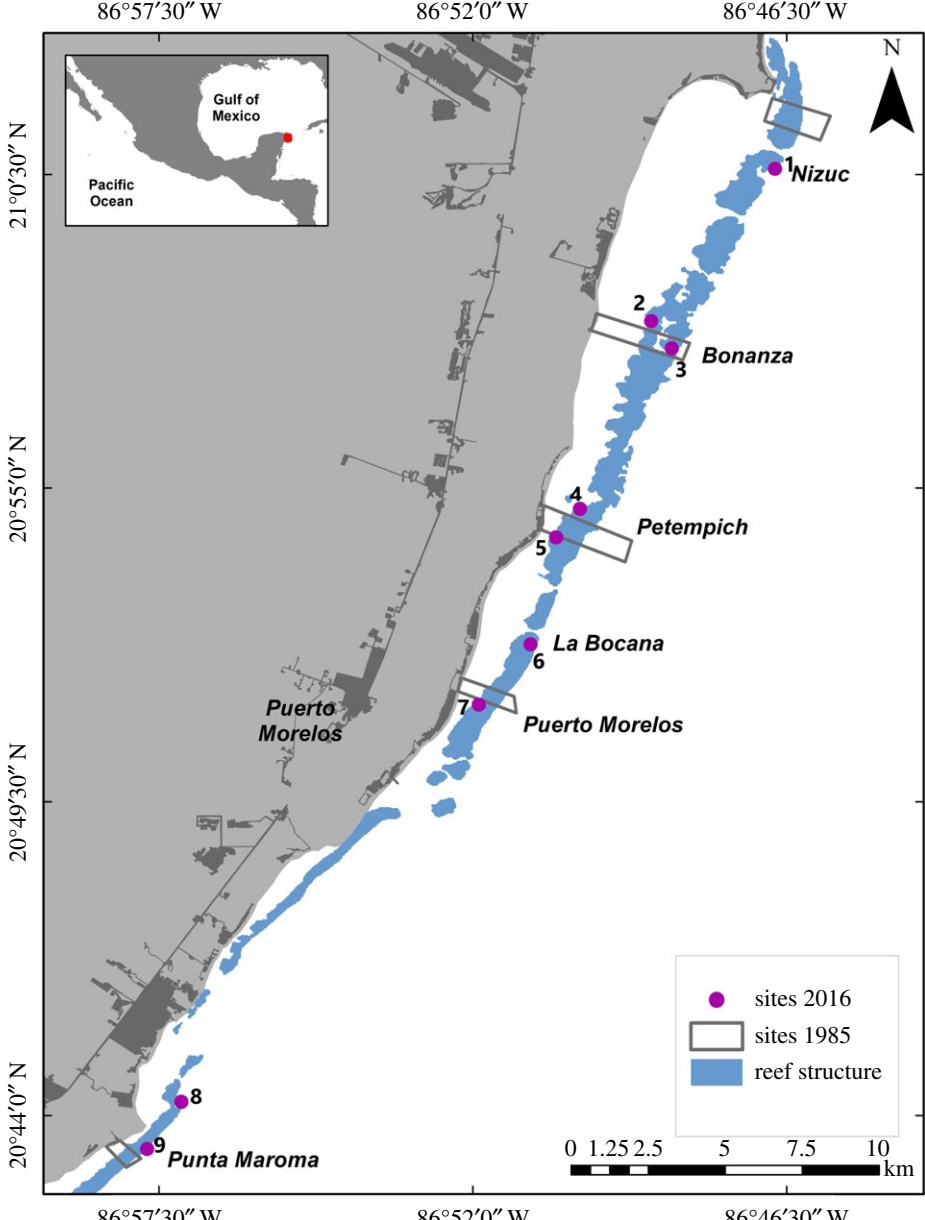

**Figure 1.** Reef sites studied in the northern sector of the Mexican Caribbean to determine changes in coral composition from 1985 to 2016. The rectangles are an approximation of the study area of the sites surveyed in 1985: Nizuc, Bonanza, Petempich, Puerto Morelos and Maroma, taken from Jordán-Dahlgren [57]. The purple circles and the numbers represent the reef sites surveyed in 2016: 1, Nizuc; 2, Bonanza; 3, Bonanza Profundo; 4, Tanchacte Norte; 5, Tanchacte Sur; 6, La Bocana; 7, Radio Pirata; 8, Punta Maroma Norte; 9, Punta Maroma Sur. The coral reefs layer is from Millennium Coral reef Mapping Project (UNEP-WCMC).

10 m strata of the fore-reef zone. We include the reef crest in the back-reef zone because the transects surveyed in 2016 extend from the back-reef to part of the reef crest within the same depth. After the reef site screening, the selected sites for study in the back-reef were Bonanza, Bocana, Petempich and Puerto Morelos and in the fore-reef zone were Bonanza, Nizuc and Punta Maroma (figure 1). The data at transect level were analysed by reef zone and used for the comparison between years. The total number of transects surveyed on the back-reef zone were 29 in 1985 and 39 in 2016, and on the fore-reef zone were 14 and 24, respectively.

### 2.2.1. Coral community changes

To explore changes in coral community composition between 1985 and 2016, we used two complementary approaches. First, we classified the coral species in four morpho-functional groups to

assess how different species with similar characteristics change overtime (electronic supplementary material, table S1). Second, we assessed broader changes in coral species composition among years and reef zones.

We distinguished four main groups of corals based on colony morphology and their contribution to reef framework [44–46]: (i) framework-building branching corals, specifically the historically important reef framework-building *Acropora* species; (ii) massive species from the second important group of reef-framework species, in this group, *Orbicella* is the main reef-builder genus, but we decided to included it with the other massive species because the contribution of *Orbicella* to the overall coral cover is relatively low; (iii) small non-framework builder species, which some authors define as opportunistic, which are small species that do not contribute greatly to calcification nor to structural complexity [44,45] and (iv) foliose-digitiform species (*Agaricia tenuifolia* and *Porites porites*), which are considered as part of the opportunistic group by some authors [44,46]. We decided to treat them as separate groups because of their differing contributions to reef three-dimensional structure at fine-scale which create important microhabitats and are susceptible to breakage (thus generating rubble), also this group is highly represented in this zone [43,58]. The mean percentage cover was calculated for each reef zone and year, both for each coral group and for the total cover.

Variation in coral species composition among years and reef zones was investigated with non-metric multi-dimensional scaling (nMDS) based on Bray–Curtis similarities of square root transformed coral cover species data in Primer v. 6 [67]. The matrix was created with the mean coral cover by species from 1985 and 2016 at the selected sites. The cover of each coral species was used as the variable, the sites as the samples, and the years and the reef zones as factors. A two-way crossed analysis of similarities (ANOSIM) was used to test the significance of these groupings (9999 permutations), with years (1985 and 2016) and reef zones as factors. We then infer the ecological space of the coral community of each reef zone per year as the total area within a polygon delineated by the exterior points (the convex hull). The convex hull area is very susceptible to extreme data points and will generally increase with sample size even if the underlying community remains the same. Consequently, we also used standard ellipse area (SEA) as a more representative measure for comparing the coral community space between reef zones in each time period. Briefly, the standard ellipse is to bivariate data as standard deviation is to univariate data. The standard ellipse of a set of bivariate data is calculated from the variance and covariance of the two axes and contains approximately 40% of the data [68]. To compare the total area for each reef zone (i.e. back-reef, fore-reef) between years, we used the Bayesian standard ellipse area corrected for sample size (SEAc) estimated and plotted using the SIBER routine for the SIAR package in R [69] and the reef zones overlap was calculated as the proportion of SEAc overlapping [70].

### 2.2.2. Reef functional changes

To assess changes in the functional capacity of studied reefs, we first estimated coral community carbonate production expressed in kg $CaCO_3$ $m^{-2}$ $yr^{-1}$, by summing the estimated $CaCO_3$ production of each species. We then computed the reef functional index (RFI) (as described below) [45] that, in addition to the carbonate production, considers the morpho-functional attributes of each species. Finally, the net carbonate budget was calculated by subtracting an estimate of erosion from the coral production (kg $CaCO_3$ $m^{-2}$ $yr^{-1}$).

Coral calcification is generally described as the product of extension rate (cm $yr^{-1}$) and the skeletal density (g $cm^{-3}$) of the coral skeleton [71]. However, because the deposition of calcium carbonate varies according to different coral morphologies, we estimated calcification rates taking into account the morphological attributes of each species following González-Barrios & Alvarez-Filip [45]. A morphometric equation was used to estimate the calcification rate of each coral species, by accounting for the morphology (cylindrical growth, octahedron, paraboloid and hemispheric), growth, extension rate (cm $yr^{-1}$) and skeletal density (g $cm^{-3}$) for each coral species. By considering the characteristics of each species (morphology and growth), potential overestimations of calcium carbonate production are avoided. The modified estimates of calcification rates calculated here represent the contribution of habitat-forming species to carbonate accumulation. Rates of coral calcification are also dependant on local environmental conditions (such as light, depth or temperature [72]); we, however, did not account for this source of variability in our analyses as local-scale information (e.g. skeletal density, growth rates) are not available for most of the coral species in our study site. To evaluate the coral carbonate production, coral cover of each species for each transect was multiplied by the calcification rate of each coral species for each reef zone and each sampling

year [45]. Data on crustose coralline algae were not available for the 1985 surveys; however, they are typically minor components [51] and therefore were omitted.

The RFI is a method proposed to estimate the species-specific functional contribution of Caribbean corals according to their capacity to create complex three-dimensional structures by means of calcium carbonate precipitation and their morphological complexity [45]. One of the main contributions of this approximation is that it provides mean estimates of calcification rate, rugosity and size for the most common Caribbean corals [45]. Here, we used these estimates, in addition to the observed cover of species, to calculate the RFI as follows. The mean estimates of calcification rate, rugosity and size of each species were scaled using the minimum and maximum value of each variable as: $X = (x - \min \text{value})/(\max \text{value} - \min \text{value})$, where $x$ is the value for each variable of each species. This standardization allows variables to have equal ranges (0–1). Then the three standardized variables were averaged to obtain a species-specific functional coefficient (Fc). The RFI is obtained through the fourth root of the summation of the product between live coral cover and the Fc of each species by transect for each reef zone and each sampling year [45]. The RFI values range from 0 to 1, where the value close to 1 represents an absolute dominance of one or several species with a highest calcification rates and the highest values of structural complexity [45].

Net carbonate production was determined as the balance between the coral carbonate production and bioerosion rates. Due to the absence of data on bioeroders from 1985, we assumed that bioerosion rates were similar in both periods of time. The rationale for this is that populations of the main reef-bioeroders have changed little during the timespan of our study. This assumption is supported by the following lines of evidence. First, our historical data (1985) were collected soon after the Caribbean-wide die-off of *Diadema antillarum* (1983–1984) [73–75]. This suggests that bioerosion rates were minimal at that time [76], because *D. antillarum* commonly accounted for up to 75% of the total bioerosion on many reefs in the region [77]. The earliest surveys of sea urchins in Puerto Morelos are from 1996 and report very low-density estimates for *D. antillarum* (0.003 ind m$^{-2}$) and for *Echinometra* spp. (0 ind m$^{-2}$) [78]. The density estimates we obtained for 2016 are slightly higher for both species (*D. antillarum* = 0.06 ind m$^{-2}$; *Echinometra* spp. = 0.03 ind m$^{-2}$; see also electronic supplementary material, table S5). Second, regarding parrotfish bioerosion, recent evidence shows that parrotfish populations in the Mexican Caribbean have undergone a slight recovery due to management regulations [7,79,80], which suggest that bioerosion followed a similar path. To further explore this, we used unpublished data on parrotfish abundance and size collected from eight sites in 2007 by the Puerto Morelos Marine Park Authority (Puerto Morelos Marine Park Authority, 2007, unpublished data). A comparison of the 2007 estimates with those obtained in 2016 confirmed that parrotfish bioerosion has slightly increased in our study area—at least during the last 10 years (electronic supplementary material, figure S2). Third, it has been predicted that the biomass and erosion rates of boring organisms (e.g. clinoid sponges) are likely to increase under ocean warming and acidification, as they will gain competitive advantages in more extreme conditions [81–83]. In sum, available evidence suggests that, in our study area, bioerosion rates may be higher than those in 1985, but since these cannot be well constrained, we have used similar rates for both periods. At worst, this would suggest our estimates of past (1985) net carbonate budgets are conservative (i.e. slightly underestimating net carbonate production).

Rates of bioerosion for 2016 were based on the assessments undertaken at the same study sites in 2017/2018 using the ReefBudget methodology [51] for the back-reef and fore-reef. Briefly, the method consisted of estimating rates of erosion by different bioeroder groups: macroborers (clinoid sponges), sea urchins, parrotfish and microborers. The area covered by individual colonies of bioeroding sponges (cm$^2$) was determined by using a transparent $5 \times 5$ cm grid, within an area encompassing 0.5 m$^2$ either side of belt-transects of $10 \times 1$ m. From this, percentage surface area covered by different sponge species can be determined. To estimate sponge bioerosion rates from census data, the methodology used published datasets to derive a relationship between sponge tissue cover and bioerosion rate whereby bioerosion rate = % surface area of sponge tissue/papillae × 0.0231 [51,84]. For the sea-urchins, the number and size class of urchins (to species) were collected in $10 \times 2$ m belt transects. To determine erosion rates by different species, ReefBudget uses published data on test size and erosion rate relationship, and since the bioerosion rates of *D. antillarum* and *Echinometra* spp. differ from other species, separate equations are used to calculate bioerosion rates (kg CaCO$_3$ m$^{-2}$ yr$^{-1}$) for *D. antillarum*, *Echinometra* and all 'other urchins'. For parrotfish bioerosion, visual censuses were conducted along $30 \times 2$ m belt transects ($N = 8$ transect per site), all parrotfish were recorded at species-level, life-phase and size. Bioerosion rate was estimated using the ReefBudget equations which use a model based on the individual size as a predictor of the amount of eroded carbonate per bite [51]. Due to the difficulties in establishing microborer rates, the ReefBudget method

uses the Caribbean rate data of Vogel *et al.* [85] for sites between 0 and 10 m depth at a rate of 0.27 kg $CaCO_3$ $m^{-2}$ $yr^{-1}$.

### 2.2.3. Data analysis

Differences between years for each reef zone among all variables (coral cover of the species groups, coral carbonate production, RFI and net carbonate production) were tested by means of non-parametric Mann–Whitney *U*-tests as the data were not normally distributed, which was verified using the Shapiro–Wilk test using the package *stats* v. 3.5 from R [86].

# 3. Results

## 3.1. Coral community changes

From 1985 to 2016, the mean coral cover in the back-reef zone showed a significant decrease from 32.90 ± 9.39% to 16.71 ± 3.55% (mean ± 95% confidence intervals (CI); Mann–Whitney *U*-test, $p < 0.05$, value tests are in electronic supplementary material, table S2). Conversely, coral cover on the fore-reef did not change significantly (1985: 13.28 ± 6.31%; 2016: 16.79 ± 3.99%; Mann–Whitney *U*-test, $p > 0.05$, electronic supplementary material, figure S1, and table S3). The proportion of cover by the four coral groups did change in both zones, but did so in different ways. In the back-reef, the cover of framework-building branching corals (i.e. *Acropora* spp.) decreased significantly from 14.16 ± 9.96% to 0.07 ± 0.15% (Mann–Whitney *U*-test, $p < 0.01$; figure 2*a*), which largely explains the overall decrease in coral cover observed in this zone. The mean cover of foliose-digitiform species also decreased significantly from 5.77 ± 3.37 to 0.84 ± 0.86% (Mann–Whitney *U*-test, $p < 0.01$) in the back-reef zone. The two species that are included in this group, *P. porites* and *A. tenuifolia*, showed similar declines for this zone. By contrast, the cover of non-framework builder species did not change significantly during the last 31 years in the back-reef (4.03 ± 1.74 to 2.66 ± 1.11; Mann–Whitney *U*-test, $p > 0.05$), and nor did cover of massive species (8.93 ± 3.32% to 13.12 ± 3.72%; Mann–Whitney *U*-test, $p > 0.05$). The apparent increase in massive species is due to the small increases in the cover of *Orbicella annularis*, *Pseudodiploria strigosa* and *Siderastrea siderea*. In the fore-reef zone, the cover of framework-building branching species (0.85 ± 0.94% to 0.45 ± 0.42%), foliose-digitiform species (0.16 ± 0.19% to 1.29 ± 1.02%) and massive species (9.69 ± 5.20% to 10 ± 3.46%) also did not change over time (Mann–Whitney *U*-test, $p > 0.05$), but the cover of non-framework species increased from 2.57 ± 1.20% to 5.04 ± 1.46% (figure 2*b*), especially *A. agaricites* and *P. astreoides*, being statistically significant between years (Mann–Whitney *U*-test, $p < 0.05$).

Coral community composition in the back- and fore-reefs zones in 1985 displayed a more scattered distribution, while that in 2016 is more tightly grouped, indicating a more similar composition between reef zones in 2016 (figure 3). A two-way crossed ANOSIM showed significant differences between sampling years across reef zone groups ($R = 0.624$, $p > 0.05$), as well as differences between reef zone groups across years ($R = 0.338$, $p = 0.05$), although with some overlap. The width (space occupied by the community) of the coral community composition was compared between years and between reef zones from the same year with the standard ellipse area (SEAc), which is the measure of the space occupied by the community (see Material and methods). The back-reef zone in 1985 had the largest SEAc of all the reef zones, in contrast with the fore-reef zone of 2016 which has the smallest SEAc (table 1). Within reef zones across years, the only overlap between SEAc of the coral community was between reef zones of 2016, with much less overlap (17%) between the SEAc of the back-reef and the SEAc fore-reef in 2016 than the other way around (39%) (table 1 and figure 3). In addition, the SEAc of the communities was smaller in both reef zones in 2016, compared with 1985, as the coral communities become more similar between reef zones.

## 3.2. Reef functional changes

Coral carbonate production decreased significantly (Mann–Whitney *U*-test, $p < 0.05$, value tests are in electronic supplementary material, table S4) in the back-reef zone between 1985 (3.51 ± 1.66 G = kg $CaCO_3$ $m^{-2}$ $yr^{-1}$), mean ± 95% confidence intervals (CI) and 2016 (1.38 ± 0.38 G) (figure 4*a*). Conversely, in the fore-reef zone, no significant differences (Mann–Whitney *U*-test, $p > 0.05$) were recorded in coral carbonate production rates between years (1985: 0.97 ± 0.51 G; 2016: 1.19 ± 0.33 G).

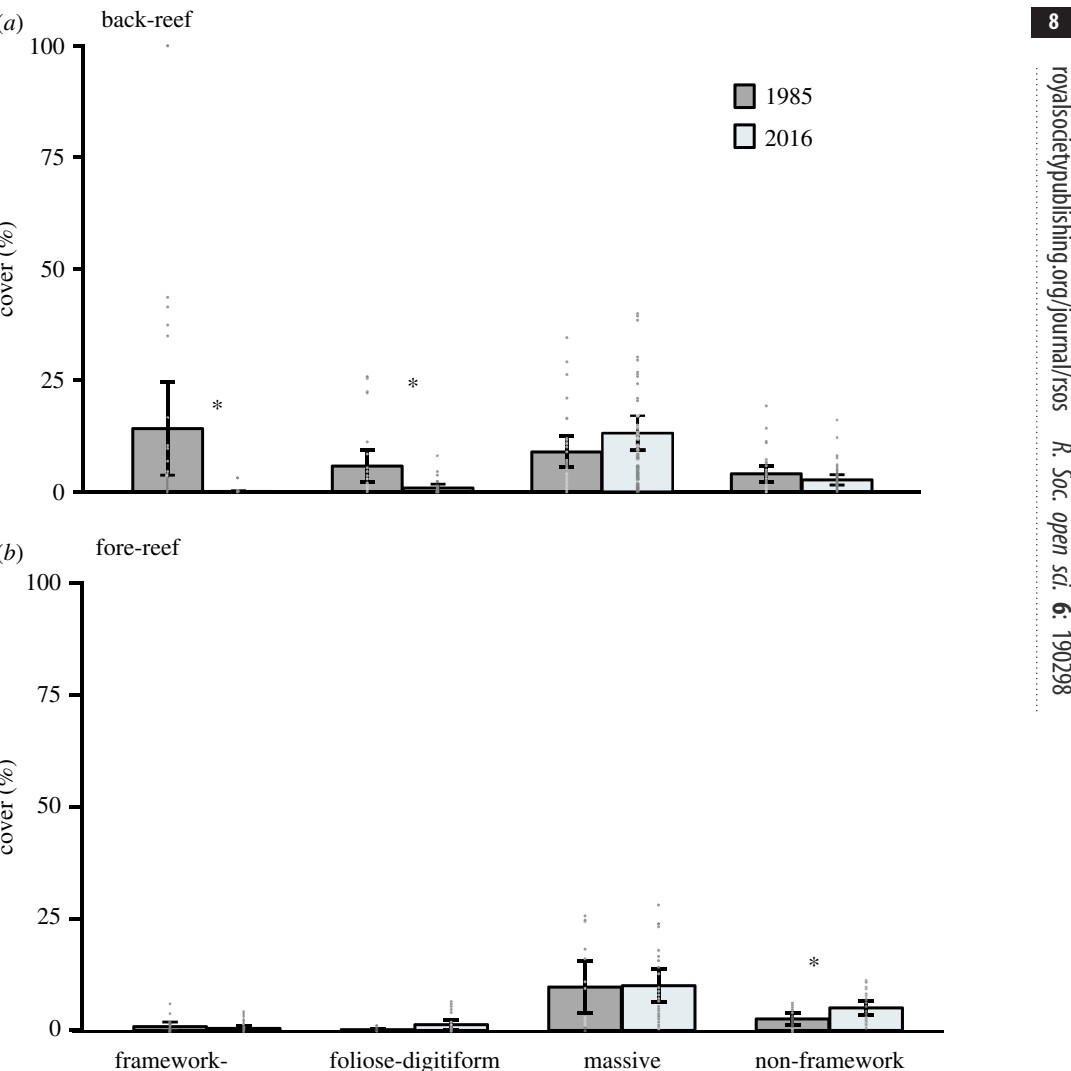

**Figure 2.** Coral cover of the different coral groups for the studied reef-zones during two sampling periods (1985 and 2016): (*a*) back-reef and (*b*) fore-reef zone. The error bars are the confidence intervals at 95% from the mean. The stars above the error bars are for the significant changes between years, and the points along the bars correspond to the data at transect level.

The RFI for the back-reef indicates a loss in the contribution of cover, structural complexity and calcification of coral species across the study period, declining from $0.57 \pm 0.66$ (mean $\pm$ CI) to $0.44 \pm 0.04$, and the drop between years was statistically significant (Mann–Whitney $U$-test, $p < 0.01$) (figure 4*b*). By contrast, no significant changes occurred in the fore-reef zone between years (1985: $0.40 \pm 0.06$; 2016: $0.45 \pm 0.03$, Mann–Whitney $U$, $p > 0.05$).

Regarding the carbonate budgets, we found, as expected, that the main carbonate producers were framework-building branching species followed by massive species, while parrotfishes accounted for most of the bioerosion in 2016. Electronic supplementary material, table S5 provides a detailed summary of the rates of erosion and production of each group. We found that in 2016, the back-reef had negative carbonate budgets, as net carbonate production dropped significantly (Mann–Whitney $U$, $p < 0.05$) from 1985 ($-0.16 \pm 1.66$ G) to 2016 ($-2.30 \pm 0.38$ G) (figure 4*c*). Although, in 1985, net carbonate production was slightly negative on average in the back-reef zone, the variation between transects is very large due to the high cover of *Acropora* in some transects, while in other transects coral cover was already very low (figure 2*a*). By contrast, only two transects of the back-reef zone had a positive, but near-neutral, carbonate budget in 2016 (figure 4*c*). In the fore-reef, the mean net carbonate budget was negative and significantly different from zero for 1985 ($-0.58 \pm 0.51$ G) and 2016 ($-0.36 \pm 0.33$ G) with most of the transects in a negative carbonate budget for both years (figure 4*c*). For the fore-reef, the net carbonate budget was not statistically different between years (Mann–Whitney $U$, $p > 0.05$).

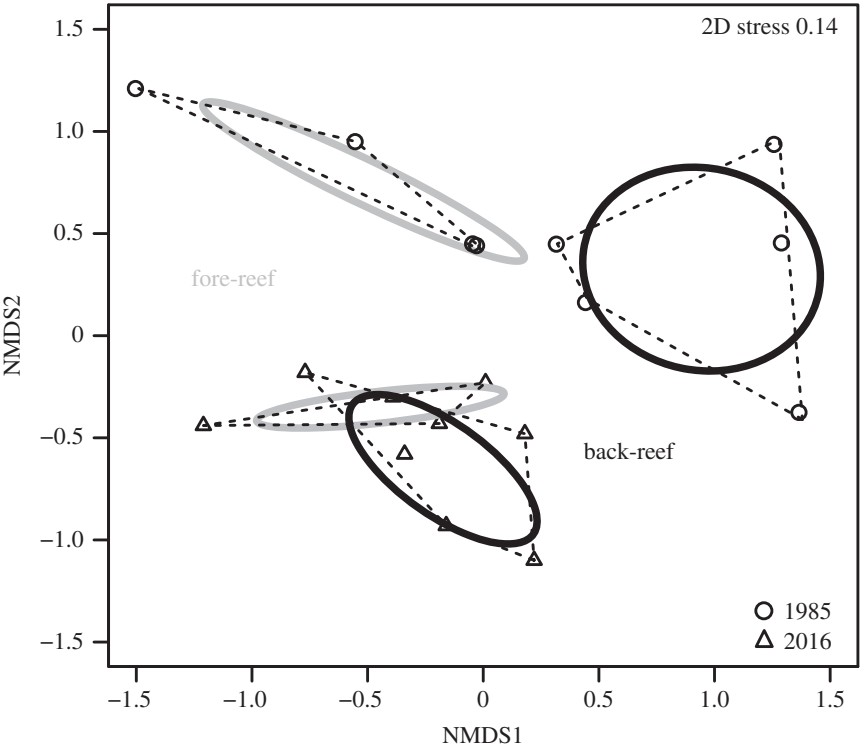

**Figure 3.** Coral community composition of the back- and fore-reef zones for the studied sites between sampling years. Non-metric multi-dimensional scaling analysis displaying degree of similarity of the community composition across 18 sites in the northern part of the Mexican Caribbean for the coral cover by species. The circles represent the sites from 1985 and the triangles the ones from 2016. The black colour stands for the back-reef zone and the grey one for the fore-reef. Dotted lines: convex hull total area (TA). Solid lines: standard ellipse area corrected for small sample sizes (SEAc).

**Table 1.** Convex hull total area (TA), Bayesian SEA, Bayesian-corrected estimate of the standard ellipse area (SEAc), overlap in SEAc between reef zones for each year and percentage of overlap with SEAc of the reef zone between years and within the same year.

| year | reef zones | convex hull total area units | SEA units | SEAc units | SEAc overlap units (%) |
|---|---|---|---|---|---|
| 1985 | back-reef | 0.77 | 0.80 | 1.07 | 0 (0%) |
| 2016 | | 0.38 | 0.34 | 0.45 | 0 (0%) |
| 1985 | fore-reef | 0.18 | 0.26 | 0.39 | 0 (0%) |
| 2016 | | 0.10 | 0.13 | 0.20 | 0 (0%) |
| 1985 | back-reef versus fore-reef | 0.77 | 0.80 | 1.07 | 0 (0%) |
| | fore-reef versus back-reef | 0.18 | 0.26 | 0.39 | 0 (0%) |
| 2016 | back-reef versus fore-reef | 0.38 | 0.34 | 0.45 | 0.8 (17%) |
| | fore-reef versus back-reef | 0.10 | 0.13 | 0.20 | 0.8 (39%) |

## 4. Discussion

Coral communities in the northern Mexican Caribbean have changed rapidly over the last three decades, leading to a structural and functional convergence of the back-reef and the fore-reef zones. In the back-reef, coral cover declined by almost 50%, largely driven by the significant loss of framework-building branching, foliose and digitiform coral species; coral cover in the fore-reef remained relatively stable despite the significant increase in non-framework building coral species (*A. agaricites* and *P. astreoides*). The increase in these non-framework species had no measurable effect on the functional potential of

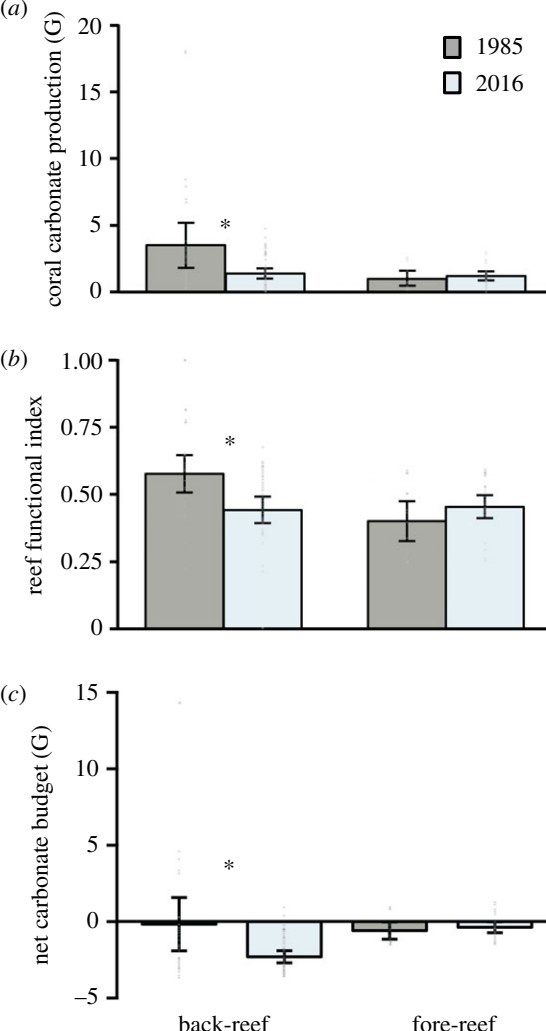

**Figure 4.** Reef function results for the back-reef and fore-reef for the studied sites between sampling years. (*a*) Rates of coral carbonate production (G). (*b*) The functional reef index which considers the morpho-functional attributes of each species. (*c*) Net carbonate production (G) estimated for both reef zones. The error bars are the 95% confidence intervals from the mean. The stars above the error bars are for the significant changes between years, and the points along the bars correspond to the data at transect level.

the fore-reefs as these species contribute very little to the reef structure and carbonate production [45]. This ecological convergence towards the dominance of low-relief species will increasingly compromise the maintenance of reef-structure and the functional potential of the reef systems in the northern part of the Mexican Caribbean; reefs in this region are now defined by negative net carbonate budgets largely determined by the presence of bioeroding organisms rather than the contribution of carbonate producers.

Environmental gradients determine the distribution and dominance patterns of coral species across the reef profiles [87–89], and on many occasions, the identity of those species defined the structural integrity and ecological complexity of the zones. For example, many reef crests across the Caribbean were historically shaped by the complex framework-building branching structures of *A. palmata* [56,90,91], and after the significant decline of this species, the genus *Orbicella* remained as the major reef-builder [8,76,92]. Unfortunately, however, populations of this taxa have decreased in the last two decades mainly due to diseases and bleaching impacts [31–34], and currently are rapidly succumbing to a recent emergent disease outbreak [35,37]. But, what happens when the coral assemblages are changing? Our study suggests that coral communities in the different reef zones changed in a non-random fashion. Specifically, major declines occurred in the important framework-building coral taxa (i.e. *Acropora*) that were the most important in delineating the back-reef zone (figure 2). Conversely, an increase in non-framework species defined the main changes observed in the fore-reef zone (figure 2).

The modification of coral communities we report here has led to a biological homogenization between reef zones, whereby instead of a dominance of reef-building coral species, there are more non-framework species that cannot fulfil the same functions as reef-builders, leaving an important niche vacant. This type of homogenization has also been observed in southern Florida, where the loss of massive-framework species led to a biotic homogenization within different locations, across depths and zones [93]. Along with the loss of massive-framework species, climatic factors are also changing the coral community assemblages, by facilitating species-range expansions into higher latitudes. *Acropora cervicornis* has, for example, moved northwards of its previously known extant range along the Florida reef tract, and this has been associated with decadal-scale increases in annual sea-surface temperatures [94]. Besides the homogenization observed across depths (i.e. [93]) and zones (as in this study), there has also been a homogenization along a latitudinal gradient. In this case, the loss of rare coral species and a potential distributional shift northwards of coral species have contributed to the homogenization in response to major disturbances like bleaching events [95].

The increase in non-framework species, especially in the fore-reef zone, is also transforming habitat configuration ([96], figure 4). Our results show that these novel reef assemblages, with simplified reef communities, that do not contribute greatly to the structural complexity, nor the carbonate budgets of reefs, can alter ecosystem functioning and productivity, as they are defined by low coral carbonate production rates, and have led to negative carbonate budgets (figure 4c). In a wider regional context, González-Barrios & Alvarez-Filip [45] found a similar situation for the rest of the Mesoamerican Reef System, where most reef-sites were considered as 'functional impaired', defined as sites with low coral cover estimates and a dominance of non-framework-building corals. Across the Mesoamerican Reef species with low reef-building potential (i.e. *Agaricia* spp. and *P. astreoides*) currently are widely dominant, while species with high functioning potential, such as *Orbicella* spp. and *Acropora* spp., have a limited relative abundance and distribution [45]. It is important, however, to recognize that within these novel assemblages with relatively low-functional potential, some species have rapid calcification rates (e.g. [95]); however, it is unlikely that these species will be capable of compensating for the loss of robust structurally complex corals given their fragile morphology; for example, in Bonaire and Curacao, the relative increase in *Madracis mirabilis* over the last four decades has compensated for some of the reduction in production observed on the reefs, but did not compensate the loss of structural complexity provided by other large calcifying species [97].

The net loss of potential to accumulate $CaCO_3$ reported here compromises the ability of coral reefs to sustain high rates of reef accretion, especially in the back-reef, which was previously the best developed zone in the north section of the Mexican Caribbean due to the contribution of the genus *Acropora* [56,80]. Gross carbonate production estimates from shallow water Caribbean reefs, before recent changes in reef ecology, are reported to have been between 10 and 17 G [98]; coral carbonate production on the back-reef in 1985 is calculated as having already been well below this (3.51 ± 1.66), but similar to levels from some of the better sites in the Caribbean measured in recent studies (i.e. greater than 4 kg; figure 4; [8,44,99]). This suggests that reefs in the Mexican Caribbean had already shifted towards net negative (and thus potentially net erosional) states before the start of our study period, a transition also suggested in recent work from Florida [10], although this needs to be taken conservatively in our case as we assumed that bioerosion rates in 1985 and 2016 were similar (see Material and methods). By contrast, by 2016, coral carbonate production was fairly negative, resembling the pathway being followed by the large majority of Caribbean reefs, where carbonate budgets tend to be neutral or negative [44]. The fore-reef sites in our study historically did not have proper reef development (see Material and methods); therefore, the carbonate production has remained low since the 1980s. Generally, shallow reefs have higher accretion rates, compared with deep reef environments [8]. However, if the loss of important structural species continues, the functionality of these environments could become more alike, especially in terms of their budget states. This idea has recently been proposed [52], and the data presented here would provide strong support for the notion that progressive depth-homogenization is occurring as a consequence of the shifting patterns of coral dominance. It is also important to remark that ocean acidification and warming may enhance destructive processes in the near future, for instance favouring the proliferation of bioeroding endolithic organisms [100], while negatively affecting coral calcification and reef-building [101].

The ecological and functional changes observed in our study sites are likely the result of ecological decline driven both by various regional-scale factors, like coral disease and bleaching, and exacerbated by local factors such as the explosive coastal development that the northern Mexican Caribbean has experienced since the onset of this study [7,37,102]. For example, local land-based threats can be synergistic with other stressors, like nutrient enrichment that increases coral susceptibility to bleaching [103]. In addition to

anthropogenic impacts, other factors like major hurricanes could be an important cause of coral loss, especially for the species with branching or foliose-digitiform morphology, since they are prone to mechanical breakage, which can lead to a decline in their coral cover [104,105]. On the other hand, low-intensity tropical storms are known to regulate the system, cooling the water or cleaning the reef bottom, leaving available substrate for coral recruitment [106].

Although the functional potential (i.e. coral carbonate production, contribution to reef-framework complexity) of many Caribbean reefs has declined over the last 4 decades; there are still sites with abundant colonies of important reef-building corals that create complex reefs and where carbonate production is greater than estimated bioerosion [30]. These 'reef oases' (sensu [107]) could be considered areas of conservation interest, due to their ability to resist disturbances and by having a coral community composition that supports the potential of positive net carbonate production. This is the case of *Limones reef*, a back-reef site, located within the study area between Nizuc and Bonanza (figure 1), but that was not part of the study because of the lack of historical data.

This site has a high cover of *A. palmata* (greater than 30%) [30], and a current estimated net production rate of 9.9 G. In addition, there is evidence suggesting that the populations of *A. palmata* in this site are highly resilient. In 2005, the cover of *A. palmata* in Limones reef dropped to less than 10% after the impact of two major hurricanes, but took less than a decade for the cover of this species to recover to its current state [30]. Improving our understanding of the mechanistic drivers underlying the persistence of sites like *Limones Reef* will be crucial to aid management and restoration efforts on our study sites and elsewhere in the Caribbean region.

The composition of coral assemblages seems to be the most important driver of the functioning of coral reefs, therefore maintaining keystone coral species could enhance the future of coral reefs [52]. If current deterioration continues, it may be expected that in the long term, non-framework coral species could also disappear since no coral species appear to be effectively insensitive to anthropogenic impact, and especially to coral disease outbreaks [37,49]. This new condition could favour the growth of other types of fauna (e.g. macroalgae, sponges, cyanobacteria) that could replace coral assemblages in the future [108]. In a rapidly changing climate, where environmental conditions are constantly modified, reefs with intermediate health and dominated by non-framework corals may thus become the new norm [109], increasing the transition away from high historical carbonate budget states to states of low net positive or negative production. Actions to address these transitions are thus urgently needed to facilitate the maintenance of the key functions that reefs provide [92], such as sand supply, vertical reef accretion and maintenance of the macro- and micro-scale framework structures that create diverse habitat space and which support many of the key reef ecosystem services provided to society.

Ethics. This study was conducted with the permission and support of the Parque Nacional Arrecife de Puerto Morelos (PROCER/CCER/PNAPM/01/2016).

Data accessibility. Details of the study including the classification of the coral species can be found in electronic supplementary material, table S1.

Authors' contributions. N.E.-S. conceived of the study with L.A.-F.; L.A.-F., E.J.-D. and N.E.-S. collected the data; N.E.-S. carried out the analyses and led the manuscript with L.A.-F., C.T.P., R.E.R.-M. and E.J.-D. All authors gave final approval for publication.

Competing interests. We declare we have no competing interests.

Funding. This study was funded by a Royal Society' Newton Advanced Fellowship (grant no. NA150360) to L.A.-F. and C.P; the Mexican Council of Science and Technology (Project PCCNCNA-055096, awarded to E.J.-D.; and the PDC-247104 warded to L.A.-F.; and the Universidad Nacional Autónoma de México (Program UNAM-DGAPA-PAPIIT, project IN-205019 awarded to L.A.-F.). N.E.-S. was supported by an MSc scholarship (no. 619930) from the CONACyT.

Acknowledgements. We thank the Postgraduate Studies in Ocean Sciences and Limnology, UNAM for the opportunity given to study her Master. We are grateful to E. Perez-Cervantes, A. González-Posada, I. Cruz-Ortega, J. González-Barrios, A. Medina-Valmaseda, E. Navarro-Espinoza, L.M. Guzmán-Fernández and A. González de la Parra who assisted greatly with data collection and logistics. We are grateful to A. Molina-Hernández whose comments and suggestions greatly improved the manuscript.

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
