## [Reviewer comments · Royal Society Open Science]

Review History

RSOS-190298.R0 (Original submission)

Review form: Reviewer 1

Is the manuscript scientifically sound in its present form?

No

Are the interpretations and conclusions justified by the results?

Yes

Is the language acceptable?

Yes

Is it clear how to access all supporting data?

No

Do you have any ethical concerns with this paper?

No

Have you any concerns about statistical analyses in this paper?

No

Recommendation?

Major revision is needed (please make suggestions in comments)

Comments to the Author(s)

See attached word document (Appendix A).

Review form: Reviewer 2

Is the manuscript scientifically sound in its present form?

Yes

Are the interpretations and conclusions justified by the results?

Yes

Is the language acceptable?

Yes

Is it clear how to access all supporting data?

No

Do you have any ethical concerns with this paper?

No

Have you any concerns about statistical analyses in this paper?

No

Recommendation?

Major revision is needed (please make suggestions in comments)

Comments to the Author(s)

Please see my comments in the document attached (Appendix B).

Review form: Reviewer 3 (Carrie Manfrino)

Is the manuscript scientifically sound in its present form?

Yes

Are the interpretations and conclusions justified by the results?

Yes

Is the language acceptable?

Yes

Is it clear how to access all supporting data?

Yes

Do you have any ethical concerns with this paper?

No

Have you any concerns about statistical analyses in this paper?

I do not feel qualified to assess the statistics

Recommendation?

Accept with minor revision (please list in comments)

Comments to the Author(s)

Please see the attached file (Appendix C).

Decision letter (RSOS-190298.R0)

19-Jun-2019

Dear Dr Alvarez-Filip,

The editors assigned to your paper ("Functional consequences of the long-term decline of reef-building corals in the Caribbean") have now received comments from reviewers. We would like you to revise your paper in accordance with the referee and Associate Editor suggestions which can be found below (not including confidential reports to the Editor). Please note this decision does not guarantee eventual acceptance.

Please submit a copy of your revised paper before 12-Jul-2019. Please note that the revision deadline will expire at 00.00am on this date. If we do not hear from you within this time then it will be assumed that the paper has been withdrawn. In exceptional circumstances, extensions may be possible if agreed with the Editorial Office in advance. We do not allow multiple rounds of revision so we urge you to make every effort to fully address all of the comments at this stage. If deemed necessary by the Editors, your manuscript will be sent back to one or more of the original reviewers for assessment. If the original reviewers are not available, we may invite new reviewers.

- Data accessibility

If you wish to submit your supporting data or code to Dryad (<http://datadryad.org/>), or modify your current submission to dryad, please use the following link:
<http://datadryad.org/submit?journalID=RSOS&manu=RSOS-190298>

- Competing interests

- Authors' contributions

- Acknowledgements

- Funding statement

on behalf of Kevin Padian (Subject Editor)
openscience@royalsociety.org

Comments to Author:

Reviewers' Comments to Author:
Reviewer: 1

Comments to the Author(s)
See attached word document

Reviewer: 2

Comments to the Author(s)
Please see my comments in the document attached.

Reviewer: 3

Comments to the Author(s)
Please see the attached file.

Author's Response to Decision Letter for (RSOS-190298.R0)

See Appendix D.

RSOS-190298.R1 (Revision)

Review form: Reviewer 1

Is the manuscript scientifically sound in its present form?
Yes

Are the interpretations and conclusions justified by the results?

Yes

Is the language acceptable?

Yes

Do you have any ethical concerns with this paper?

No

Have you any concerns about statistical analyses in this paper?

No

Recommendation?

Accept with minor revision (please list in comments)

Comments to the Author(s)

Dear editor, dear authors

This is the second time I review this manuscript and I am very pleased with all the changes I don't see any major flaws in the updated version and the authors did a very good job addressing and changing all the point I raised before. I am happy to suggest publication of this manuscript after the final minor points as elaborated on below are addressed.

Minor comments:

Line 35: Point after *Acropora* spp.

Line 48: What exactly do you mean with "ecological performance", consider elaborating a bit more on what specific aspects of reef ecology are lost

Line 52: Space in front of i.e.

Line 58: With few I assume you mean a few species, not that there were not many framework building corals? It is not clear as it is currently written

Line 69: consider removing "resultant"

Line 77: I would be careful with such a statement. Predicting what will happen within reef communities is near impossible. I would use words like "...we can hypothesize about future reef assemblages..."

Line 79: is this indeed the ultimate consequence? Consider re-wording

Line 113: "and descends gradually to a depth of..." this last section seems to not flow from the first part. Consider splitting.

Line 126: what about (artificial) beaches in this region? Are there any?

Line 146: Counted? Or was the cover under the line measured?

Line 151: consider removing 'percentage'

Line 163: were analysed

Line 178: "...but we decided to included it with? the...."

Line 296: spp. After *Echinometra*

Line 300: Does the reefbudget method not include more parrotfish species?

Line 325: might this have anything to do with the placement of the transects? How was the placement in 2016 chosen? Randomly? If not it could be possible that transect lines were placed on more developed reef sections, while some parts of the reef that used to be dominated by massive species may now have been transformed to sand or rubble patches.

Line 340: be consistent with the spaces in your representation of $R=0.6$ and $R=0.3$

Line 365: consider changing to "...we found, as expected,"

Line 370: it would be useful to present observed ranges of net carbonate production next to the average production of the different zones.

Line 402: Not only a recent outbreak also previous ones and bleaching as you mentioned already in the introduction.

Line 411: Space = niche?

Line 431: consider replacing the - sign by a comma.

Line 434: No point behind spp and astreoides not in italic.

Line 435:have a limited....

Line 437: recognized should be recognize

Line 454: use either - signs or comma's to break this sentence, not both as in this case.

Line 459: fore-reefs sites, remove the s of reefs

Line 467: should be enhance

Line 470: space between building and [

Line 587: our recent publication (Extreme spatial heterogeneity in carbonate accretion potential on a Caribbean fringing reef linked to local human disturbance gradients accepted in GCB and in collaboration with Chris Perry) might be of interest regarding local variation in carbonate budgets and reef oases.

Review form: Reviewer 2

Is the manuscript scientifically sound in its present form?

Yes

Are the interpretations and conclusions justified by the results?

Yes

Is the language acceptable?

Yes

Do you have any ethical concerns with this paper?

No

Have you any concerns about statistical analyses in this paper?

Yes

Recommendation?

Accept with minor revision (please list in comments)

Comments to the Author(s)

The clarity and organization of the manuscript is much improved in the revision and it is obvious that the authors carefully considered the suggestions of myself and the other reviewers. In particular, I appreciate the additional information about historic and present-day bioerosion in the study area, which justified their choice to use 2016 bioerosion rates for both modern and historical carbonate budgets. The flow of the Discussion has also been improved substantially and in general, the language of the manuscript is much more clear throughout the manuscript. There are a few places where the text could still use some rephrasing, which I have outlined in the Minor points below. Overall, I suggest that the manuscript be accepted to ROCS after some additional minor revisions.

The only significant concern I have, which I overlooked in the first review of the manuscript relates to the data analysis described in Lines 305-309. I'm wondering why the data weren't analyzed with a two-way test (i.e., a Kruskal Wallis test or an ANOVA with the data transformed

or ranked) analogous to the two-way ANOSIM that was used to analyze the multivariate data? The major results of these analyses are likely robust, but testing separately for the effects of time and zones is not appropriate because these variables are not independent.

Minor suggestions

Line 22: I'd change to "the last several decades" or just "recent decades"

Line 23-24: Add a comma after species and add "and" before leading

Line 34: Add period after spp.

Line 36: It might be good to add examples of the non-framework species whose abundance has increased as well.

Line 41: sustain should be "sustains"

Line 71: In the abstract you include *A. tenifolia* as an example of a framework-building species. I would either change *Agaricia* spp. here to be more specific about what species are non-framework-builders (*A. agaricites*?) or just take it out and leave the *P. astreoides* example.

Line 99: Add the closed parentheses after "non-framework"

Line 110: add a comma after Historically

Line 113: add a comma after development and change "descends" to "descended"

Line 128: change "in average" to "on average"

Line 132: add a comma after ref. 56

Line 134: change "coral reefs distribution" to either "coral-reef distribution" or "coral reefs' distribution"

Line 149: add a comma after zone and after eight

Line 151: change "at" to "to"

Lines 170-173: It's not clear to me what the difference between the first and second analysis is based on this sentence. Is the first sentence referring to the functional group analysis? If so, make that clear.

Line 177-180: I would simplify this sentence to something like: massive framework-building species, primarily *Orbicella* spp. as other massive taxa had low cover at our sites.

Line 181: I would add "which" before "are" to keep parallel structure

Lines 183-184: Similarly, I would change "these species" to "which" and start a new sentence with "we decided"

Line 196: Perhaps change "width" to something like ecological space?

Line 229: change "of" to "on"

Line 230: need a closed parenthesis after reference 70

Line 247: change "of" to "for"

Line 252: change "the" to "a"

Line 319: *Acropora* needs to be italicized

Lines 321-324: is this also in the back-reef zone? Please make this clear.

Line 323: "The two species..." should be a separate sentence

Line 324: I'd suggest changing "Contrary" to something like "In contrast"

Line 325: "significant" should be "significantly"

Line 333: I would also add a list of what species actually increased in abundance in parentheses (*P. astreoides* and *A. agaricites*?)

Line 343: add a comma after (SEAc)

Line 350: I would consider changing to something like: was smaller in both reef zones in 2016 compared with 1985, as the coral communities became more similar across reef zones. Because "getting smaller" implies that the change is continuous and ongoing, which is not clear from your study.

Line 367: change "provided" to "provides"

Line 374-375: This is not a complete sentence as currently written. Rephrase to something like: In contrast, only two transects of the back-reef zone has a positive, near neutral carbonate budget in 2016.

Line 377: change “which was” to “and the change between years”. Also, are the carbonate budgets in the fore-reef significantly negative or does the uncertainty overlap with zero. Please make this clear.

Line 409: I’d suggest changing “builder” to “building”

Line 419: Change the Burman et al. reference to its reference number

Line 422: Change “cases” to “case”

Line 425: You’re missing a space in Fig.4

Line 428: Add “have” before “led”

Line 433: You’ve used *Agaricia* rather than *Undaria* elsewhere. I think the community is back to using *Agaricia* now, right?

Line 434: *astreoides* should be italicized

Line 437: change “recognized” to “recognize”

Line 452: I’d say “reefs in the Mexican Caribbean had...” for clarity

Line 456: I’d remove “In contrast” from the beginning of this sentence since the next sentence starts the same way

Line 479: Something is wrong with the wording here. Perhaps “since they are prone to mechanical breakage, which can lead to a decline in their cover.”

Lines 480-482: The language of this sentence is also not clear. May “On the other hand, low intensity storms can also have positive impacts on reefs, by cooling...”

Line 487: Change “consider” to “considered”

Line 496: Is “actual” the right word here? Natural, maybe? Also please change “Improve” to “Improving”

Figures 2 and 4 captions: In the last sentence “de” should be “the”

Figure 4: in the caption and in the text you say carbonate production, but the axis label is Coral calcification rates, which really isn’t accurate. In the caption for b) I’d suggest saying “, which considers...” for parallel structure.

Table 1: what does the 2 superscript indicate?

Decision letter (RSOS-190298.R1)

03-Sep-2019

Dear Dr Alvarez-Filip:

On behalf of the Editors, I am pleased to inform you that your Manuscript RSOS-190298.R1 entitled "Functional consequences of the long-term decline of reef-building corals in the Caribbean" has been accepted for publication in Royal Society Open Science subject to minor revision in accordance with the referee suggestions. Please find the referees' comments at the end of this email.

The reviewers and Subject Editor have recommended publication, but also suggest some minor revisions to your manuscript. Therefore, I invite you to respond to the comments and revise your manuscript.

- Ethics statement

- Data accessibility

If you wish to submit your supporting data or code to Dryad (<http://datadryad.org/>), or modify your current submission to dryad, please use the following link:
<http://datadryad.org/submit?journalID=RSOS&manu=RSOS-190298.R1>

- Competing interests

- Authors' contributions

- Acknowledgements

- Funding statement

Because the schedule for publication is very tight, it is a condition of publication that you submit the revised version of your manuscript before 12-Sep-2019. Please note that the revision deadline will expire at 00.00am on this date. If you do not think you will be able to meet this date please let me know immediately.

on behalf of Prof Kevin Padian (Subject Editor)
openscience@royalsociety.org

Associate Editor Comments to Author:

The reviewers consider your manuscript much-improved. There are a number of concerns remaining, however, and these need to be addressed in your revision - in particular, we'd draw

your attention to the concerns of reviewer 2 regarding the statistical tests applied. Please ensure you respond fully to the queries of the referees in your revision - good luck!

Reviewer comments to Author:

Reviewer: 2

Comments to the Author(s)

The clarity and organization of the manuscript is much improved in the revision and it is obvious that the authors carefully considered the suggestions of myself and the other reviewers. In particular, I appreciate the additional information about historic and present-day bioerosion in the study area, which justified their choice to use 2016 bioerosion rates for both modern and historical carbonate budgets. The flow of the Discussion has also been improved substantially and in general, the language of the manuscript is much more clear throughout the manuscript. There are a few places where the text could still use some rephrasing, which I have outlined in the Minor points below. Overall, I suggest that the manuscript be accepted to ROCS after some additional minor revisions.

The only significant concern I have, which I overlooked in the first review of the manuscript relates to the data analysis described in Lines 305-309. I'm wondering why the data weren't analyzed with a two-way test (i.e., a Kruskal Wallis test or an ANOVA with the data transformed or ranked) analogous to the two-way ANOSIM that was used to analyze the multivariate data? The major results of these analyses are likely robust, but testing separately for the effects of time and zones is not appropriate because these variables are not independent.

Minor suggestions

Line 22: I'd change to "the last several decades" or just "recent decades"

Line 23-24: Add a comma after species and add "and" before leading

Line 34: Add period after spp.

Line 36: It might be good to add examples of the non-framework species whose abundance has increased as well.

Line 41: sustain should be "sustains"

Line 71: In the abstract you include *A. tenifolia* as an example of a framework-building species. I would either change *Agaricia* spp. here to be more specific about what species are non-framework-builders (*A. agaricites*?) or just take it out and leave the *P. astreoides* example.

Line 99: Add the closed parentheses after "non-framework"

Line 110: add a comma after Historically

Line 113: add a comma after development and change "descends" to "descended"

Line 128: change "in average" to "on average"

Line 132: add a comma after ref. 56

Line 134: change "coral reefs distribution" to either "coral-reef distribution" or "coral reefs' distribution"

Line 149: add a comma after zone and after eight

Line 151: change "at" to "to"

Lines 170-173: It's not clear to me what the difference between the first and second analysis is based on this sentence. Is the first sentence referring to the functional group analysis? If so, make that clear.

Line 177-180: I would simplify this sentence to something like: massive framework-building species, primarily *Orbicella* spp. as other massive taxa had low cover at our sites.

Line 181: I would add "which" before "are" to keep parallel structure

Lines 183-184: Similarly, I would change "these species" to "which" and start a new sentence with "we decided"

Line 196: Perhaps change "width" to something like ecological space?

Line 229: change “of” to “on”

Line 230: need a closed parenthesis after reference 70

Line 247: change “of” to “for”

Line 252: change “the” to “a”

Line 319: *Acropora* needs to be italicized

Lines 321-324: is this also in the back-reef zone? Please make this clear.

Line 323: “The two species...” should be a separate sentence

Line 324: I’d suggest changing “Contrary” to something like “In contrast”

Line 325: “significant” should be “significantly”

Line 333: I would also add a list of what species actually increased in abundance in parentheses (*P. astreoides* and *A. agaricites*?)

Line 343: add a comma after (SEAc)

Line 350: I would consider changing to something like: was smaller in both reef zones in 2016 compared with 1985, as the coral communities became more similar across reef zones. Because “getting smaller” implies that the change is continuous and ongoing, which is not clear from your study.

Line 367: change “provided” to “provides”

Line 374-375: This is not a complete sentence as currently written. Rephrase to something like: In contrast, only two transects of the back-reef zone has a positive, near neutral carbonate budget in 2016.

Line 377: change “which was” to “and the change between years”. Also, are the carbonate budgets in the fore-reef significantly negative or does the uncertainty overlap with zero. Please make this clear.

Line 409: I’d suggest changing “builder” to “building”

Line 419: Change the Burman et al. reference to its reference number

Line 422: Change “cases” to “case”

Line 425: You’re missing a space in Fig.4

Line 428: Add “have” before “led”

Line 433: You’ve used *Agaricia* rather than *Undaria* elsewhere. I think the community is back to using *Agaricia* now, right?

Line 434: *astreoides* should be italicized

Line 437: change “recognized” to “recognize”

Line 452: I’d say “reefs in the Mexican Caribbean had...” for clarity

Line 456: I’d remove “In contrast” from the beginning of this sentence since the next sentence starts the same way

Line 479: Something is wrong with the wording here. Perhaps “since they are prone to mechanical breakage, which can lead to a decline in their cover.”

Lines 480-482: The language of this sentence is also not clear. May “On the other hand, low intensity storms can also have positive impacts on reefs, by cooling...”

Line 487: Change “consider” to “considered”

Line 496: Is “actual” the right word here? Natural, maybe? Also please change “Improve” to “Improving”

Figures 2 and 4 captions: In the last sentence “de” should be “the”

Figure 4: in the caption and in the text you say carbonate production, but the axis label is Coral calcification rates, which really isn’t accurate. In the caption for b) I’d suggest saying “, which considers...” for parallel structure.

Table 1: what does the 2 superscript indicate?

Reviewer: 1

Comments to the Author(s)

Dear editor, dear authors

This is the second time I review this manuscript and I am very pleased with all the changes I don't see any major flaws in the updated version and the authors did a very good job addressing

and changing all the point I raised before. I am happy to suggest publication of this manuscript after the final minor points as elaborated on below are addressed.

Minor comments:

Line 35: Point after *Acropora* spp.

Line 48: What exactly do you mean with “ecological performance”, consider elaborating a bit more on what specific aspects of reef ecology are lost

Line 52: Space in front of i.e.

Line 58: With few I assume you mean a few species, not that there were not many framework building corals? It is not clear as it is currently written

Line 69: consider removing “resultant”

Line 77: I would be careful with such a statement. Predicting what will happen within reef communities is near impossible. I would use words like “...we can hypothesize about future reef assemblages...”

Line 79: is this indeed the ultimate consequence? Consider re-wording

Line 113: “and descends gradually to a depth of...” this last section seems to not flow from the first part. Consider splitting.

Line 126: what about (artificial) beaches in this region? Are there any?

Line 146: Counted? Or was the cover under the line measured?

Line 151: consider removing ‘percentage’

Line 163: were analysed

Line 178: “...but we decided to included it with? the....”

Line 296: spp. After *Echinometra*

Line 300: Does the reefbudget method not include more parrotfish species?

Line 325: might this have anything to do with the placement of the transects? How was the placement in 2016 chosen? Randomly? If not it could be possible that transect lines were placed on more developed reef sections, while some parts of the reef that used to be dominated by massive species may now have been transformed to sand or rubble patches.

Line 340: be consistent with the spaces in your representation of $R = 0.6$ and $R = 0.3$

Line 365: consider changing towe found, as expected,

Line 370: it would be useful to present observed ranges of net carbonate production next to the average production of the different zones.

Line 402: Not only a recent outbreak also previous ones and bleaching as you mentioned already in the introduction.

Line 411: Space = niche?

Line 431: consider replacing the - sign by a comma.

Line 434: No point behind spp and *astreoides* not in italic.

Line 435:have a limited....

Line 437: recognized should be recognize

Line 454: use either - signs or comma's to break this sentence, not both as in this case.

Line 459: fore-reefs sites, remove the s of reefs

Line 467: should be enhance

Line 470: space between building and [

Line 587: our recent publication (Extreme spatial heterogeneity in carbonate accretion potential on a Caribbean fringing reef linked to local human disturbance gradients accepted in GCB and in collaboration with Chris Perry) might be of interest regarding local variation in carbonate budgets and reef oases.

Author's Response to Decision Letter for (RSOS-190298.R1)

See Appendix E.

Decision letter (RSOS-190298.R2)

23-Sep-2019

Dear Dr Alvarez-Filip,

I am pleased to inform you that your manuscript entitled "Functional consequences of the long-term decline of reef-building corals in the Caribbean: evidence of across-reef functional convergence" is now accepted for publication in Royal Society Open Science.

on behalf of Prof Kevin Padian (Subject Editor)
openscience@royalsociety.org

Appendix A

This manuscript describes changes in the coral assemblage on the Mexican reef Puerto Morelos since 1985 and the repercussions these changes have on gross and net carbonate production and reef structural complexity. The authors have looked at sites within the back-reef and fore-reef zone and especially in the former the coral community has changed considerably. Compared to 1985 branching (mostly *Acropora* spp.) and foliose coral species have strongly declined in cover and non-framework building species have become a proportionally important group among corals. The authors state that both the relative shift in groups within the coral community as well as the general decline in coral cover has strongly impacted the capacity of these reefs to maintain net carbonate production. Although I appreciate the work conducted by the authors and the challenge they faced comparing novel data to incomplete historical data (e.g. no information on bioerosion) I have a few concerns about the presented work.

The authors describe the change in the coral community based on major groups distinguished by various life-history traits. The authors only haphazardly describe some changes in specific coral species. Although it makes sense to look at functional groups for some analyses and comparisons I think a lot of information is lost by not looking at the individual coral species as well. In the branching group it is clear that the loss of Acroporid corals has resulted in a considerable decline, but now it is unclear how, for instance, *Orbicella* spp. have changed. On the back-reef there appears to be a small, albeit not significant, increase in cover of massive species. It would be very interesting to find out the massive species found in 2016 compare to 1985. I would maybe expect more (*pseudo*)*diploria* spp. The same goes for the non-framework building corals, do you find the same species in 1985 and 2016? This would also be relevant in light of the described homogenization, maybe this is true for groups but not at all for specific important species.

Somewhat in line with this issue I think it is necessary to provide a more detailed description of how these reefs looked in 1985 and even before. The authors touch upon a number of relevant issues associated with reef degradation, including the loss of positive net growth, structural complexity and shifts within the coral community. Although all these changes indeed impact reef functioning throughout the Caribbean region they seem less relevant to the reefs described in this manuscript. If I understood correctly, net reef carbonate production was, on average, already negative in 1985. This would mean that since that time reefs have been losing structural complexity? I think it would be very useful to provide some information on the carbonate production and erosion factors. Was there for instance a lot of erosion in 1985? And if so, by which organisms? I am surprised to see there is such low carbonate production if there used to still be a considerable cover of *Acropora* species which are known to have fast rates of carbonate production. Also, it is now unclear if the authors had rugosity data for 1985. A short section describing the historical reef morphology, reef community, etc. of these reefs would help to place the current findings in a historical context.

The authors point out that they found considerable variation in carbonate production both in 1985 and 2016 among sites. This implies a lot of variation on the studied reef stretch. If this is the case, I am wondering if the authors surveyed enough reef sites to come to a representable mean. I realize that the number of sites is limited by the sites surveyed in 1985, but maybe some more justification is needed. I would also recommend to use boxplots in figure 4 and also show the original data points in it. This would give a much clearer view of the spread within the data. It would maybe be interesting to show the individual transects as point. Within the text it could be better to give ranges with the averages that the authors provide, especially for carbonate production, which can differ considerably between transects/sites. Following up on this the authors should consider including a supplementary table with values for carbonate production (maybe separated by functional group), estimated erosion by group of organisms, rugosity, etc. on a transect level.

Overall, the manuscript often does not read very smooth. Especially in the introduction and discussion, it seems the authors jump from thought to thought where more explanation is needed (e.g. page 3 line 17). I feel that especially the discussion could do with a better structure. The authors should make a clear distinction between the findings of this study and findings described in previous literature, right now this is not always very clear what their novel contribution is. Normally I don't think a general comment like this is very useful so I will try to give some examples:

"The increase of these non-framework species had no measurable effect on the functional potential of the fore-reefs as these species contribute very little to the reef structure and carbonate production [36]."

"This modification of coral communities has led to a biologically homogenization between reef zones, whereby instead of a dominance of reef-builder coral species (a situation which was both historically and geologically the norm in the Caribbean; [18,81,82]), there are more non-framework species that cannot fulfil the same functions as reef-builders, leaving an important space vacant."

In part I think this has to do with the build up of sentences. They are often very long and in multiple occasions the start of the sentence does not link properly to the end. Here are some examples of difficult sentences and some in which I feel the cause-effect is not fully correct in my eyes:

"The resultant decline of the major reef-building coral species across the Caribbean has led to a relative increase in the abundance of non-framework coral species, such as *Agaricia* spp. and *Porites astreoides* [32–34]." Is the decline in RB species indeed the main cause for the increase in non-framework building species?

"The ultimate consequence of a reduced abundance of important reef-building species will reduced reef-carbonate budgets, with rates of bioerosion becoming increasingly important controls on overall budget states" An ultimate consequence is not that bioerosion becomes increasingly important. Consider splitting the sentences.

"By considering the characteristics of each species (morphology and growth), potential overestimations of calcium carbonate production are avoided, and thus represent the contribution of habitat forming species to carbonate accumulation."

As a general point I wonder what the effect of declined coral cover is on the issues described in this manuscript. The main focus now lies on the shift in coral communities. And although this indeed has a clear effect, but figure S1 shows an overall decline in cover of about 50% in the back reef. I think that this has a significant impact on for instance carbonate production as well. Yet this is not really covered in detail in the discussion.

Minor comments by line. I apologize for any inconsistencies regarding line numbering. In the document I received line numbers did often not align with the actual sentence.

Page 2

Line 15: 'function' may on itself be a bit vague.

Line 18: shortly mention how the coral communities were evaluated (method)

Line 24: *Acropora* in italic (see more occasions throughout the manuscript)

Line 33: it feels like the last sentence should be linked to the previous or a verb needs to be added

Page 3

Line 18: compromising should be compromises

Line 27: do hurricanes not also cause physical erosion?

Line 47: Mainly disease? Bleaching through thermal stress also had a major effect I assume

Page 4

Line 14: reduce (and space after).

Line 16: example of a sentence that could be more to the point: for instance: *will transition into states of net erosion*

Line 21: Maybe personal, but *reef structure* could mean a lot of thing, structural complexity?

Line 41: are the groups really only distinguished by colony shape?

Line 45: What exactly do the authors mean with physical reef functioning

Page 5

Line 8: consider including a lead sentence introducing the study site Puerto Morelos, now it comes a bit out of nowhere

Line 10: Is it both a fringing reef and a barrier reef?

Line 15: what do the authors mean here with "mostly associated"

Line 16: "*Historically it has a well-developed back-reef and reef-crest that were dominated by Acropora palmata*" What about *cervicornis*? I am just interested to know. Also, what does this mean for carbonate production? It seems that a reef dominated by *palmata* should have a considerable positive net production. I think here you could use the known literature to describe a bit more the reefs at Puerto Morelos, also historically.

Line 28: 'in the coast'  in the coastal area?

Line 30: this sentence is a bit long and becomes confusing, also could the authors provide some more detail on the sewage status? Where does this go? Does it really all seep in? what about run-off? Untreated discharge through pipes?

Line 56: here the max depth is 20 m, but before the authors talk about 30 m

Line 57: It is not entirely clear how the transect where placed. Where 20m transects placed in each zone? Or did a transect cover multiple zones since they were placed perpendicular? Also, how haphazardly were transects placed in both years? How was the start point determined?

Page 6

Line 9: the applied methods in 1985 and 2016 are very different. If there is enough data I think it is oke to compare, but the authors should be very careful with drawing strong conclusions especially because it seems there is considerable variation within the reef. Maybe it would be good to also mention the potential shortcomings of this in the discussion?

Line 17: I understand they may not have collected GPS data, but it was not pre-GPS. Formulate a bit different, maybe: ... before the general use of GPS in scientific research...

Line 25: why only these two zones?

Line 35, remove comma before and.

Figure 1

- Font of coordinates is too small

- Why is site 1 located so far from sites in 1985?

- I don't think symbology is the correct word

Page 7

Line 52: it is not clear why the authors go from classifying species to persistent through time

Line 55: this part I miss a bit in the results, the specific changes in specific coral species

Page 8

Line 3: Only *Acropora*? What about *Orbicella*?

Line 35: be careful throughout the manuscript not to use too many unnecessary abbreviations and if abbreviations are used be consistent.

Page 9

Line 5: space between units

Line 26: estimated **and expressed** in units?

Line 32: I partly agree, but the calcification by CCA should not be underestimated, especially in the cryptic environment and in 1985. If the cover was measured you could consider including it or mention that it was actually low in both years but then provide a value

Line 37: is there rugosity data for 1985?

Line 37: example of abbreviation (this is the first mention of RFI right?)

Line 39: how were the data standardized?

Line 42: the following section needs some more elaboration, for the reader it is not fully clear what applies to the RFI or the Fc.

Line 47: not clear what the authors mean here with 4th root. Was the data transformed? I was wondering about this in general. Did the authors consider transforming the data?

Page 10

Line 12: Space between Caribbean and refs

Line 24: abundant that  abundant than

Line 35: I realize it is impossible to retrieve data on bioerosion, but this section includes a lot of assumptions. What about excavating sponges? They can be important eroders, especially on Caribbean reefs and their abundance increased in recent years.

Line 52: The value for sponge erosion is outdated. The reef budget method at present also includes species specific rates for sponge erosion. Otherwise also see De Bakker et al. 2018, PlosOne.

Page 11

Line 10: I thought more species specific rates were available

Line 14: which constants were used? 0-5 or 5-10 for the different zones? Maybe also point out the approximate depth range of the two studied zones earlier.

In the coral community section it would be very interesting to include some species specific information. What reef builders do now dominate the back-reef? Are they the same as in 1985? More *Diploria*?

Page 12

Line 52/52: a general remark, consider presenting statistical characters in italic and be consistent with space in presented p values (e.g. $p = 0.01$). See for instance also Page 14 line 34, here the authors use = and <.

Figure 3:

- No purple or green colour, I only see grey and black
- write out MBRS

Page 13

Line 13: make sure it is clear what kind of homogenization you are talking about, among sites, within sites, among zones. It will not always be correct to talk about homogenization since it seems that there is a lot of variation still between sites and the actual contribution of the various groups compared to the relative contribution.

Page 14

In the section starting at line 40 it is very hard to understand what the authors want to say. Also it seems that the sentences are not correctly written.

Line 45: here the authors should mention why it was already negative and that there was a lot of variation in 1985. This is a strange finding without explanation and could question the relevance of this study. If the reef was already disappearing then what has changed?

Line 51: still significant yes but the authors don't claim it changed significantly, so why the 'but....'

Figure 4:

- Consider the use of boxplot with the actual data points
- explain what the authors mean with 'height'
- I assume these are **95%** CI
- explain what the stars mean (I assume significant change?) Also in figure 2

Page 16

Line 15: This seems somewhat strange, how was this effect measured? I am sure that a decline in rugosity while cover remains the same (with different species) could be measured. If not, I wonder if the authors can make this claim based on their data.

Line 24: is there evidence in the data that this will happen?

Line 25: they were already negative in 1985. This sentence implies they are currently in a negative state but were not before. Also I am not sure if it is correct to make a claim on a change in the impact of bioerosion because bioerosion was assumed to be similar in both years.

Page 17

Line 5: *Along with the loss of massive-framework species, climatic factors are changing the coral community assemblages.* Is the loss of massive species not also the result of climate change and local impact?

Line 9: I wonder if the move northwards is really a relevant issue here, north of the studied reefs the *Acropora*'s have seen massive mortality as well.

Line 14: does this apply to the Florida reef or the data collected for this study? And here it is not clear what kind of homogenization is meant. Are we now talking about homogenization on a species level?

Line 20: the crown-of-thorns-starfish is not an issue in the Caribbean right?

Line 23: is the spatial variation based on species composition of on the groups.

Line 23: I don't see why spatial variation on itself is responsible for the potential loss of carbonate production

Line 27: elaborate on the meaning of well developed.

Line 37: I don't think this is what it suggests.

Line 42: What do the authors mean by neutral?

Line 43: fore-reefs

Line 45: How historically? Is there data from before 1985. Has there never been a well developed reef there?

Line 57: additional issue

Page 18

Line 3: Yes, but what about *Orbicella* spp. in your data?

Line 11: Yes, but is this an issue of your reefs if they have never had net positive carbonate production

Line 12: yes, the rubble is a point, but is it relevant to your study? Now it comes a bit out of nowhere.

Line 15-19: True, but again this seems not too relevant for the reefs described in this study because they were already negative.

Line 30: I agree these kind of reefs can serve as a buffer, but maybe the authors should elaborate a bit on the reasons why. E.g. recruitment, etc.

Line 47: The issue that the community will not go back is more related to the idea that the impact on reefs nowadays will not likely become better, rather it will likely become worse.

Line 49: what do the authors mean with simplified communities? A community dominated by *acropora* corals is in my opinion more simple than a community with many different organisms.

Page 19

Line 19: Is nutrient enrichment not also caused by large by the absence of functional waste water treatment? Other stressors are more temperature, sedimentation, etc....

Line 24: *overwhelming the available space that will eventually overgrow corals.* This sentence needs to be changed now it seems the available space is overgrowing corals.

Line 30: **composition of** the coral assemblage

Line 39: and vice versa, other organisms can also cause decline or recovery of reef-builders.

I would like to point out that I believe this work describing temporal changes in reef carbonate production potential can be a valuable contribution when aforementioned concerns are addressed in an appropriate way.

Appendix B

Review of Estrada-Saldivar et al.

This paper provides a long-term perspective on the changes in community assemblages and reef function on coral reefs in the Mexican Caribbean. The retroactive analysis of historical data presented in this manuscript provides much-needed context for understanding the modern decline of reef ecosystems and will be an important contribution to the literature. The authors consider these data using a variety of ecological and geological metrics, which allows for a more nuanced assessment of how the reef has changed over the last 30 years, compared with more traditional studies that only consider changes in coral cover.

Many of my comments are fairly minor suggestions about the language of the manuscript (outlined under Minor Comments), which I think will improve the clarity and flow of the text. I do, however, have a few more substantive comments/questions that I would like to see the authors address as well, which I have outlined below:

- (1) P5, L19-25: The impact of hurricanes on these reefs is an important point, that the authors should revisit in the Discussion: could this be an important cause of the loss of branching and foliose-digitiform corals in the back-reef environments?
- (2) P5, L50/51: Regarding the differences in the methodologies used to quantify percent cover of coral taxa. I agree with the statement that percent cover estimates by the two methods are likely similar, but I'm curious how you think estimated carbonate production using chain-based methods (which measure the surface area of the colony) may differ from the AGRRA method where the line is pulled taut. More rugose colonies would have higher estimated carbonate production with the former method, correct? Could this exaggerate the differences between the surveys? I think it would be good to add a sentence or to acknowledging this potential source of uncertainty.
- (3) P8, L38-41 & P12, L54-60: When discussing the TA and SEA analyses, I would rephrase the language to describe what these metrics actually mean ecologically. For example, it's not clear to me what "width area of the mean coral community" means.
- (4) P9, L53-P10, L33: I think it's fair to say that *Diadema* played only a minor role in bioerosion throughout the study given that their populations declined dramatically in the early 1980s, but the role of parrotfish (and potentially other urchins) in bioerosion in the past is a source of uncertainty in your study that you need to acknowledge head-on. There is no evidence that parrotfish populations in the MBRS were "clearly low" in the 1980s. Jackson's study suggests that there was fishing pressure in the Caribbean for a long time, but there is little direct evidence that parrotfish populations were low Caribbean-wide at this time. Importantly, none of these studies cited here provide any information about parrotfish populations in Mexico in the 1980s, so we really don't know whether there were changes during your study period. Similarly, while there is clear evidence that *Diadema* populations were already low at the beginning of your study

period, there is no data on *Echinometra*, which are significant contributors to bioerosion on parts of the MBRS. Relatedly, I don't think that you have enough evidence to conclude that your estimated carbonate budgets are conservatively low. Without clear evidence to the contrary, it is just as likely, that in many places, the impact of parrotfish on bioerosion has declined as their populations have declined, as Perry et al. showed for Bonaire.

I think it's fine to state here that you were simply not able to make inferences about changes in bioerosion due to changing parrotfish populations and then, discuss that point of uncertainty further in the Discussion. Another option would be to do a sensitivity analysis that compares carbonate production with, for example, a doubling and a halving of parrotfish populations relative to modern surveys. This could potentially allow you to determine how much uncertainty is represented by the lack of past parrotfish data.

- (5) P11, Data Analysis: Since so many analyses were done on different versions of the same dataset, you might consider performing a Bonferroni correction to decrease the probability of making a Type I error.
- (6) P12, L50-53: Is the first ANOSIM result you're reporting the reef zone effect or the interaction of the two factors? If it's the result for reef zone, it's not correct to say that there were "significant differences between sampling years" across reef zones. That result is telling you that there is a difference in community composition between zones, irrespective of year. If it's the interaction, then you should also report the reef zone effect.
- (7) P12-13, Community composition: the sentence on P13, L7-10 is very similar to the first sentence of this section. Although you're talking about two different metrics, these results say the same thing. I would consider re-organizing so that you discuss these points together, rather than talking about significant differences between years/zones in between. It's also not clear to me what the result on P13, L4-7 means. Isn't this just saying that there is more variability in the community composition of the back reef than in the fore reef?
- (8) P13, Figure 3: The figure caption refers to different colors, but the figure is in black and white. Additionally, I would add a key on the plot so that readers glancing at the figure know which years/zones are represented by which polygons. You should also report the stress of the nMDS.
- (9) P14, Reef functional changes: I think it would be good to include a summary of the data from the modern carbonate budget surveys (perhaps just in the Supplementary). This way readers can evaluate the impact of bioerosion on these reefs and which taxa contribute most significantly to this process. This will also allow you to discuss the potential changes in bioerosion between 1985 and 2016 more explicitly (see comments 4 and 11).

- (10) Discussion: I would like to suggest that the authors consider some re-organization of the Discussion to streamline its flow and minimize repetitions. First, I suggest that the authors move the section on P19, L7-28 about the causes of reef decline along the Mexican coast just after the discussion of the decline in *A. palmata* that ends at P16, L42. I would also move the text about the decline of *Orbicella* spp. here. I also suggest moving the section about novel species assemblages P18, L39-60 just after the discussion of biotic homogenization because these are similar themes rather than after discussions of the impacts of these changes (on carbonate production, complexity, function). Finally, I think that the two paragraphs on changes in carbonate production/complexity (P17, L22-P18, L37) could be streamlined and/or combined because there is some repetition in these sections (i.e., about the decline in carbonate budgets in the Caribbean).
- (11) I would also like to see some discussion of the following topics added to this section 1) role of storms in reef decline (see comment 1), 2) the RFI. This is a really interesting metric, but it's not mentioned at all in the discussion. Can you put the changes in RFI into a broader context? What do they mean about the state of the reefs in your study? How do your results compare with those of Gonzalez-Barrios and Alvarez-Filip? 3) bioerosion and a source of uncertainty in your results. You might also want to talk about the idea proposed by some researchers that bioerosion could increase in the future.
- (12) P17, L43-45: why did the fore-reef environments here historically have poor reef development?
- (13) P18, L29/30: I think you need to be more clear about what you mean by buffer sites. What is it that these reefs could be preventing/providing? Perhaps discuss how reefs that remain functional could act to sustain regional biodiversity and may be important targets for management? I think the concept of "reef oases", as presented in Guest et al. 2018, Journal of Applied Ecology could be useful to add here.
- (14) P19, L23-24: I'd be careful here. There is little evidence that macroalgae actively overgrow corals under normal circumstances. When this (rarely) happens, it's generally only small colonies, not adult corals. More typically, macroalgae abundance has been observed to increase as space is opened up on the reef after corals die.

Minor Comments

Abstract:

P2, L5: change in to on

P2, L12: insert relative before dominance

P2, L17: comma after Here

P2, L23: change In to Over and add a comma after years

P2, L24-26: *Acropora* should be italicized and there should be a period after spp.

P2, L33-35: the final sentence is a fragment and should be combined with the previous sentence

Introduction:

P3, L15/16: I would add some references at the end of the sentence about the causes of coral decline

P3, L17/18: I suggest changing loss to decline and compromising should be changed to has compromised for parallel structure

P3, L19: It's not clear to me what the phrase "and hence diversity" means. Are the authors referring to the diversity of reef-associated biota? If so, please make that clear or simply remove this parenthetical

P3, L21 (and throughout): there should be a hyphen in sea-level rise

P3, L22/23; i.a. should be i.e. You could also cite Toth et al. 2018. Global Change Biology here.

P3, L28: I would break up the references, so it is clear which support the phrase about erosion/OA and which refer to hurricane impacts, since ref 7 was broken out in the first part of the sentence. You could also cite the newly published study by Kuffner et al. (2019, Limnology and Oceanography), which also discusses reef erosion on in the Florida Keys.

P3, L39: there should be a space between Pleistocene and the references

P3, L44: *Orbicella* is a single genus so, this should read: massive corals in the genus

P3, L45/46: "due" is missing after mainly

P3, L56/57: I would suggest citing at least one of Rich Aronson's studies (e.g., Aronson et al. 2004, Ecology) here because he was really the first to document increases in *Agaricia* populations following disturbance events. I would also make it clear in the next sentence that these taxa are relative dominants not the dominant benthos on the reefs in most cases.

P4, L13/14: missing the word "be" before reduced.

P4, L17: budget should be budgets

P4, L34/35: change between to in because here between suggests that surveys were done in years other than 1985 and 2016, which is not the case

Methods:

P5, L13: perhaps change associated with to determined by or driven by

P5, L14/15: change has to had

P5, L16/17: change is to was

P5, L18: change that to and

P5, L28/29: change in to on or along

P5, L30: comma before which

P5, L40: I'd change which to who

P5, L43/44: does "their distribution" mean the distribution of reef zones or the distribution of coral taxa across reef zones? Consider rephrasing.

P5, L55/56: Change on to at. I'd also add a comma after five

P6, L17/18: I'd change this to the surveys

P6, L26-30: The wording of this sentence is somewhat confusing. I'd change to something like We also include the reef crest in the back-reef zone because the transects surveyed in 2016

extend from the back reef to part of the reef crest. You may also want to reiterate that both zones are in similar depths.

P6, L33-35: It's not clear to me what the phrase "integrated into the reef zone level" means. Can you just say that transect data were summarized for each site?

P7, L37: either remove this was or make that the beginning of a separate sentence

P7, L60: I would consider making the name of this group more specific since other branching corals fall into different groups.

P8, L10: It would be useful to indicate at what size corals were considered non-framework builders instead of massives

P8, L55/56: there's an extra space between ^-2 and you're missing a space before year

P8, L57/58: I'd either add a reference to Gonzalez-Barrrios & Alvarez-Filip here or add something like "as described below" in a parenthetical.

P9, L5: to should be from

P9, L7: by should be as

P9, L9: missing a space after skeleton

P9, L23-25: the wording of the final phrase of this sentence is awkward. I'd make it a separate sentence and say something like: The modified estimates of calcification rate calculated here represent...

P9, L24-26: You've already stated the units for calcification rate twice in the previous paragraph: there is no need to repeat that here.

P9, L28: I'd change by to on each or for each

P9, L37: add (RFI) after Reef Function Index

P9, L39: I'd rephrase of each variable to for each of these metrics. Please also include a statement about how the data were standardized (by mean?)

P9, L42-46: I think it makes more sense to move this sentence to the end of the paragraph

P9, L60: need a d at the end of change

P10, L11/12: need a space after Caribbean

P11, L24: Change reef functional index to RFI or at least capitalize for consistency

P11, L27: change this to which or make a new sentence

Results

P11, L39-42: I'd add 95% before confidence intervals. Also, in some places Supplementary Table is fully capitalized and in other places table is lowercase.

P11, L48: Again, I would indicate that you're talking about framework-building branching corals here.

P11, L55: significant should be significantly

P11, L58/59: builder should be building

P12, L7: I'd change being to and was

P12, L52-54: the phrasing of this sentence is awkward. I would combine with the previous sentence: and between groups...you also need a comma before although.

P12, L59: replace smaller with smallest

P13, L40: should read non-metric multi-dimensional scaling (nMDS)

P13, L41/42: MBRS hasn't been defined anywhere in the manuscript

P14, L19-21: Why is the data presentation different here? You're reported means±CI elsewhere already.

P14, L30/31: delete actual

P14, L34: comma after In contrast

P14, L41: I'd rephrase this to: ...1985, we found that in 2016...

P14, L45-48: I'd add "on average in the back-reef zone" after negative. There were several places in the Results and Discussion where it was not immediately clear to me which reef zone you were talking about. I would also delete "given that" from this sentence.

P14, L50: You're just talking about the back reef here too right? I would add that before transects.

P15, L35: add 95% before confidence intervals

Discussion

P16, L9: add a comma after back-reef

P16, L32: change in to on

P16, L58: I'd change space to niche

P17, L14-21: You need to provide context for where this study was conducted, otherwise I think discussion of COTs, which aren't found in the Caribbean, is confusing. This is also just a very long sentence, so I would consider breaking it up.

P17, L33: I would rephrase this sentence to something like: ...back-reef at our sites in 1985 was already well below this (add G from your site here), but...

P17, L41/42: I would say production was fairly negative, not neutral in the back-reef environments by 2016.

P17, L51: perhaps change discussed to proposed

P17, L57: addition should be additional

P18, L21: hyphen in sea-level rise and erosion should be erosional

P18, L23-24: isn't loss of three-dimensional complexity the same thing as breakdown of reef structural complexity. Delete this phrase

P18, L33: comma before which and after area

P18, L45/46: need a space after conditions

P18, L52/53: you need a citation at the end of this sentence

P18, L59/60: has should not be italicized

P19, L3/4: need a space after reefs

P19, L19: add citations at the end of this sentence

P19, L12: you may want to cite other studies that have found similar results here. See Bruno et al. 2019. Climate change, coral loss, and the curious case of the parrotfish paradigm: why don't MPAs improve resilience? Annual Review of Marine Science and Cox et al. 2017. MEPS. Establishment of marine protected areas alone does not restore coral reef communities in Belize

P19, L40/41: I'd move "in the future" to the end of the sentence

P19, L42: modifying should be changed to being modified or simply changing

P19, L45/46: on coral reefs is repetitive. I'd delete. I'd also put a comma after the citation

Appendix C

Supplementary Information.

Table S1.

The CORAL TYPE COLUMN has a mixture of morphological information and function (non-framework and presumable framework). For clarity you can separate information on function and coral morphology type. Some of the species designated as Non-framework are not framework at their sites, but they are part of the framework at other Caribbean locations. (*Madracis*, *Agaricia*) are an integral part of the framework of many Caribbean reefs.

Pg 3 lines 23- 27 *The* ecological homogenization of the reef is provoking observation. The change is largely the consequence of the mortality of *Acropora* spp. so the reef has changed from having mostly large branching corals and foliose digitiform coral species in the back-reef zone.

26 add which foliose-digitiform coral species you are referring to in (parentheses).

28 Non-framework species. *Agaricia* spp. could in fact be a framework species. Consider including all of the *Agaricids* as foliose then recalculate the changes in accretion and homogenization.

30 -32 Calcification rates are decreasing. You explain nicely. Certain species that are decreasing were faster growing. Is there an overall measured decrease in calcification of species that are persisting?

Also, the % of coral cover has decline and therefore calcification and accretion (eg loss of 3D structure).

PG 8 Line 58- Clearly describes all of this is this morphological classification should be included in Table S1 for consistency.

P 10 Lines 7-27 Intraspecific calcification can be highly variable. Calcification is also a function of water depth and habitat (energy levels- eg leeward vs windward sides of islands).

P19 Line 58 “addition” do you mean “additional”

Appendix D

UNIVERSIDAD NACIONAL AUTÓNOMA DE MÉXICO INSTITUTO DE CIENCIAS DEL MAR Y LIMNOLOGÍA UNIDAD ACADÉMICA DE SISTEMAS ARRECIFALES

ADDRESS: PROL. AV. NIÑOS HEROES S/N, PUERTO. MORELOS, QR 77580 MEXICO
POSTAL ADDRESS: APARTADO POSTAL 1152, CANCÚN, QR 77500 MEXICO
TEL: 998 871 0219, 998 871 0009

26th July 2019

Dear Professor Jeremy Sanders,
Editor in chief, *Royal Society Open Science*.

We thank you and the reviewers for your comments on the manuscript '**Functional consequences of the long-term decline of reef-building corals in the Caribbean**'. The comments were very helpful and gave us the opportunity to substantially improve the manuscript considerably.

Below we provide details of how we have amended the manuscript in response to the reviewers' comments (shown in **blue**). We look forward to hearing from you and hope you will consider that following these revisions our manuscript is now suitable for *Royal Society Open Science*.

Yours faithfully,

Nuria Alejandrina Estrada Saldivar
(nuria.estradasaldivar@hotmail.com)

Dr Lorenzo Alvarez-Filip

Biodiversity and reef conservation lab (UNAM)
www.barcolab.org

Reviewer: 1

Comments to the Author

This paper provides a long-term perspective on the changes in community assemblages and reef function on coral reefs in the Mexican Caribbean. The retroactive analysis of historical data presented in this manuscript provides much-needed context for understanding the modern decline of reef ecosystems and will be an important contribution to the literature. The authors consider these data using a variety of ecological and geological metrics, which allows for a more nuanced assessment of how the reef has changed over the last 30 years, compared with more traditional studies that only consider changes in coral cover.

Many of my comments are fairly minor suggestions about the language of the manuscript (outlined under Minor Comments), which I think will improve the clarity and flow of the text. I do, however, have a few more substantive comments/questions that I would like to see the authors address as well, which I have outlined below

We thank your positive comments and below we provide a detailed description of how we have amended the manuscript following your comments and questions.

(1) P5, L19-25: The impact of hurricanes on these reefs is an important point, that the authors should revisit in the Discussion: could this be an important cause of the loss of branching and foliose-digitiform corals in the back-reef environments?

We agree and have included this on the discussion (Lines 470-475) in the resubmitted version) "...In addition to anthropogenic impacts, other factors like major hurricanes could be an important cause of coral loss, especially for the species with branching or foliose-digitiform morphology, since are really prone to break that led to a decline in their coral cover [103,104]; albeit low intensity tropical storms are known to regulate the system, cooling the water or cleaning the reef bottom, leaving available substrate for coral recruitment [105]".

(2) P5, L50/51: Regarding the differences in the methodologies used to quantify percent cover of coral taxa. I agree with the statement that percent cover estimates by the two methods are likely similar, but I'm curious how you think estimated carbonate production using chain-based methods (which measure the surface area of the colony) may differ from the AGRRA method where the line is pulled taut. More rugose colonies would have higher estimated carbonate production with the former method, correct? Could this exaggerate the differences between the surveys? I think it would be good to add a sentence or to acknowledging this potential source of uncertainty.

We agree with this rationale. However, in our approach the carbonate production of each species was directly estimated from its cover along the transect (and we did not consider the colony-level rugosity). So, because the two methods provide comparable estimates of cover the carbonate production estimates should be equally similar.

In addition, please note that to account for species morphology and complexity we used the method proposed by González-Barríos and Álvarez-Filip 2019 that provides estimates of species calcification rate considering each species' morphology and structural complexity. A detailed description of this approach is provided in the methodology section lines 235-250.

(3) P8, L38-41 & P12, L54-60: When discussing the TA and SEA analyses, I would rephrase the language to describe what these metrics actually mean ecologically. For example, it's not clear to me what "width area of the mean coral community" means.

Thank you for giving us the opportunity to improve the description of these analyses. We have rephrased the description and interpretation of these metrics as follows:

In methods (lines 195-207):

“...We then infer the width of the coral community of each reef zone per year as the total area within a polygon delineated by the exterior points (the convex hull). The convex hull area is very susceptible to extreme data points and will generally increase with sample size even if the underlying community remains the same. Consequently, we also used standard ellipse area (SEA) as a more representative measure for comparing the coral community space between reefs zones in each time period. Briefly, the standard ellipse is to bivariate data as standard deviation is to univariate data. The standard ellipse of a set of bivariate data is calculated from the variance and covariance of the two axes and contains approximately 40% of the data [67]. To compare the total area for each reef zone (i.e. back-reef, fore-reef) between years, we used the Bayesian standard ellipse area corrected for sample size (SEAc) estimated and plotted using the SIBER routine for the SIAR package in R [68] and the reef zones overlap was calculated as the proportion of SEAc overlapping [69].”

In Results (lines 338-347):

“...The width (space occupied by the community) of the coral community composition was compared between years and between reef zones from the same year with the standard ellipse area (SEAc) which is measure of the space occupied by the community (see methods). The back-reef zone in 1985 had the largest SEAc of all the reef zones, in contrast to the fore-reef zone of 2016 which has the smallest SEAc (Table 1). Within reef zones across years, the only overlap between SEAc of the coral community was between reef zones of 2016, with much less overlap (17%) between the SEAc of the back-reef and the SEAc fore-reef in 2016, than the other way around (39%) (Table 1, Fig. 3). In addition, the SEAc of the communities is getting smaller as the coral communities become more similar between reef zones.”

(4) P9, L53-P10, L33: I think it's fair to say that *Diadema* played only a minor role in bioerosion throughout the study given that their populations declined dramatically in the early 1980s, but the role of parrotfish (and potentially other urchins) in bioerosion in the past is a source of uncertainty in your study that you need to acknowledge head-on. There is no evidence that parrotfish populations in the MBRS were “clearly low” in the 1980s. Jackson's study suggests that there was fishing pressure in the Caribbean for a long time, but there is little direct evidence that parrotfish populations were low Caribbean-wide at this time. Importantly, none of these studies cited here provide any information about parrotfish populations in Mexico in the 1980s, so we really don't know whether there were changes during your study period. Similarly, while there is clear evidence that *Diadema* populations were already low at the beginning of your study period, there is no data on *Echinometra*, which are significant contributors to bioerosion on parts of the MBRS. Relatedly, I don't think that you have enough evidence to conclude that your estimated carbonate budgets are conservatively low. Without clear evidence to the contrary, it is just as likely, that in many places, the impact of parrotfish on bioerosion has declined as their populations have declined, as Perry et al. showed for Bonaire.

I think it's fine to state here that you were simply not able to make inferences about changes in bioerosion due to changing parrotfish populations and then, discuss that point of uncertainty further in the Discussion. Another option would be to do a sensitivity analysis that compares carbonate production with, for example, a doubling and a halving of parrotfish populations relative to modern surveys. This could potentially allow you to determine how much uncertainty is represented by the lack of past parrotfish data.

Thank you for these comments as they have given us the opportunity to expand our justification. We now provide more evidence to support that bioerosion rates in 2016 were similar or slightly higher than those in previous decades. First, we have included a

study that reports sea urchin density (for both *Diadema* and *Echinometra*) for Puerto Morelos in 1996; and as expected the density of these two species was slightly higher in our 2016 surveys. Second, we provide unpublished data (from 2007) about parrotfish erosion on the same study area to demonstrate that parrotfish bioerosion has indeed increased in Puerto Morelos. As suggested in another comment, we have also included a sentence about the possibility that sponge bioerosion will increase in the context of ocean acidification and global warming.

Lines 251-277

“Net carbonate production was determined as the balance between the coral carbonate production and bioerosion rates. Due to the absence data on bioeroders from 1985, we assumed that bioerosion rates were similar in both periods of time. The rationale for this is that populations of the main reef-bioeroders have changed little during the timespan of our study. This assumption is supported by the following sources of evidence. First, our historical data (1985) was collected soon after the Caribbean-wide die-off of *Diadema antillarum* (1983-1984) [73–75]. This suggests that bioerosion rates were minimal at that time [76], because *D. antillarum* commonly accounted for up to 75% of the total bioerosion on many reefs in the region [77]. The earliest surveys of sea urchins in Puerto Morelos are from 1996 and report very low-density estimates for *D. antillarum* (0.003 ind/m²) and for *Echinometra* spp (0 ind/m²) [78]. The density estimates we obtained for 2016 are slightly higher for both species (*D. antillarum* = 0.06 ind/m²; *Echinometra* spp = 0.03 ind/m²; see also Table S5). Second, regarding parrotfish bioerosion, recent evidence shows that parrotfish populations in the Mexican Caribbean have undergone a slight recovery due to management regulations [7,79,80] which suggest that bioerosion followed a similar path. To further explore this, we used *unpublished* data on parrotfish abundance and size collected from eight sites in 2007 by the Puerto Morelos Marine Park Authority. A comparison of the 2007 estimates with those obtained in 2016 confirmed that parrotfish bioerosion has slightly increased in our study area - at least during the last 10 years (Fig. S2). Third, it has been predicted that the biomass and erosion rates of boring organisms (e.g. clinoid sponges) are likely to increase under ocean warming and acidification, as they will gain competitive advantages in more extreme conditions [81–83]. In summary, available evidence suggests that, in our study area, bioerosion rates may be higher than those in 1985, but since these cannot be well constrained, we have used similar rates for both periods. At worst this would suggest our estimates of past (1985) net carbonate budget are conservative (i.e., slightly underestimating net carbonate production).”

Please also note that we have remained the reader about this in the Discussion (L 449-451) section.

(5) P11, Data Analysis: Since so many analyses were done on different version of the same dataset, you might consider performing a Bonferroni correction to decrease the probability of making a Type I error.

Although we performed the same test we did use different datasets for most of the comparisons. This is that the dataset that we used for each analysis is different; therefore we respectfully do not consider that this correction should be applied. Please, also note that Bonferroni corrections not advisable for studies with a small number of comparisons (see Streiner, 2011; Napierala, 2012; Armstrong, 2014).

(6) P12, L50-53: Is the first ANOSIM result you're reporting the reef zone effect or the interaction of the two factors? If it's the result for reef zone, it's not correct to say that there were "significant differences between sampling years" across reef zones. That result is telling you that there is a difference in community composition between zones, irrespective of year. If it's the interaction, then you should also report the reef zone effect.

We performed a Two-way crossed ANOSIM and we have reworded the results to make clear this. Lines 335-338 "... A two way crossed ANOSIM, showed significant differences between sampling years across reef zone groups ($R= 0.624$, $p >0.05$), as well as differences between reef zones groups across years ($R=0.338$, $p = 0.05$), although with some overlap."

(7) P12-13, Community composition: the sentence on P13, L7-10 is very similar to the first sentence of this section. Although you're talking about two different metrics, these results say the same thing. I would consider re-organizing so that you discuss these points together, rather than talking about significant differences between years/zones in between.

Thank you for your comment. We have deleted the second sentence.

It's also not clear to me what the result on P13, L4-7 means. Isn't this just saying that there is more variability in the community composition of the back reef than in the fore reef?

This paragraph has been reworded following your comment #3

(8) P13, Figure 3: The figure caption refers to different colors, but the figure is in black and white. Additionally, I would add a key on the plot so that readers glancing at the figure know which years/zones are represented by which polygons. You should also report the stress of the nMDS.

Thank you for your comment, the figure caption was corrected and we added the stress of the nMDS and a key on the plot to specify which years/zones are represented by which polygons.

"...The circles represent the sites from 1985 and the triangles the ones from 2016. The black colour stands for the back-reef zone and the grey one for the fore-reef. Dotted lines: convex hull total area (TA). Solid lines: standard ellipse area corrected for small sample sizes (SEAc)."

(9) P14, Reef functional changes: I think it would be good to include a summary of the data from the modern carbonate budget surveys (perhaps just in the Supplementary). This way readers can evaluate the impact of bioerosion on these reefs and which taxa contribute most significantly to this process. This will also allow you to discuss the potential changes in bioerosion between 1985 and 2016 more explicitly (see comments 4 and 11).

In the Supplementary material with now include a table with summary information of carbonate producer and bioeroders for each reef zone. Please also note that we now provide more evidence to support that bioerosion rates in 2016 were similar or slightly higher than those in previous decades, which includes a new analysis (Fig. S2).

(10) Discussion: I would like to suggest that the authors consider some re-organization of the Discussion to streamline its flow and minimize repetitions.

Thank you for giving us the opportunity to improve the flow of the Discussion. Following your comments and those from the Second Referee, we have reviewed the entire Discussion in order to provide a better structure, avoid redundancies and make a clear

distinction between the findings of our study and findings described in previous literature.

First, I suggest that the authors move the section on P19, L7-28 about the causes of reef decline along the Mexican coast just after the discussion of the decline in *A. palmata* that ends at P16, L42.

We decided to keep this paragraph towards the end of the Discussion because does not represent one of our major findings (here we only provide a list of potential causes that could explain the observed trends). We however, hope that with the new structure this paragraph fits better in this place.

I would also move the text about the decline of *Orbicella* spp. here.

We have moved the text about *Orbicella* just after the discussion of *A.* decline as suggested.

I also suggest moving the section about novel species assemblages P18, L39-60 just after the discussion of biotic homogenization because these are similar themes rather than after discussions of the impacts of these changes (on carbonate production, complexity, function).

Moved as suggested, but also please note we have reworded this paragraph.

Finally, I think that the two paragraphs on changes in carbonate production/complexity (P17, L22-P18, L37) could be streamlined and/or combined because there is some repetition in these sections (i.e., about the decline in carbonate budgets in the Caribbean).

The two paragraphs were combined see lines 439-464.

(11) I would also like to see some discussion of the following topics added to this section 1) role of storms in reef decline (see comment 1),

This was addressed as part of the discussion, as detailed in your first comment.

2) the RFI. This is a really interesting metric, but it's not mentioned at all in the discussion. Can you put the changes in RFI into a broader context? What do they mean about the state of the reefs in your study? How do your results compare with those of Gonzalez-Barrios and Alvarez-Filip?

Thank you for pointing out this opportunity. He have added the following in L 424-431: "...In a wider regional context, González-Barrios & Alvarez-Filip (2018) found a similar situation for the rest of the Mesoamerican Reef System, where most reef-sites were considered as 'functional impaired' - defined as sites with low coral cover estimates and a dominance of non-framework building corals. Across the Mesoamerican Reef a species with low reef-building potential (i.e. *Undaria* spp and *Porites astreoides*) currently are widely dominant, while species with high functioning potential such as *Orbicella* and *Acropora* had a limited relative abundance and distribution across the region [45]."

3) bioerosion and a source of uncertainty in your results. You might also want to talk about the idea proposed by some researchers that bioerosion could increase in the future.

Thank you we have added the following in L 461-464

(12) P17, L43-45: why did the fore-reef environments here historically have poor reef development?

In the section of study site, we now provide a better description of this zone. Lines 110-118:

“... Historically it had a well-developed back-reef and reef-crest that were dominated by *Acropora palmata*, which contributed greatly to the structural complexity of the reef; while the fore-reef was relatively flat, with no significant reef framework development and descends gradually to a depth of ~20-25m into an extensive sand platform [56,57]. Historically, the fore-reef zone was mostly of low relief, gentle slope and colonized by coral grounds. The most conspicuous components of the fore-reef zone were octocorals, macroalgae, and small coral heads [57].”

(13) P18, L29/30: I think you need to be clearer about what you mean by buffer sites. What is it that these reefs could be preventing/providing? Perhaps discuss how reefs that remain functional could act to sustain regional biodiversity and may be important targets for management? I think the concept of "reef oases", as presented in Guest et al. 2018, *Journal of Applied Ecology* could be useful to add here.

We have reworded the whole paragraph and included your suggestion.

L 476-491:

“Although the functional potential (i.e. coral carbonate production, contribution to reef-framework complexity) of many Caribbean reefs have declined over the last four decades; there are still sites with abundant colonies of important reef-building corals that create complex reefs and where carbonate production is greater than estimated bioerosion [30]. These “reef oases” [*sensu* 106] could be consider areas of conservation interest, due to their ability to resist disturbances and by having a coral community composition that supports the potential of positive net carbonate production. This is the case of *Limones reef* (see Fig. 1), a back reef site, located within the study area, but that was not part of the study because of the lack of historical data. This site has a high cover of *A. palmata* (>30%) [30], and a current estimated net production rate of 9.9 G. In addition, there is evidence suggesting that the populations of *A. palmata* in this site are highly resilient. In 2005 the cover of *A. palmata* in Limones reef dropped to less than 10% after the impact of two major hurricanes but took less than a decade for the cover of this species to recover to its actual state [30]. Improve our understanding of the mechanistic drivers underlying persistence of sites like *Limones Reef* will be crucial to aid management and restoration efforts on our study sites and elsewhere in the Caribbean region.”

(14) P19, L23-24: I'd be careful here. There is little evidence that macroalgae actively overgrow corals under normal circumstances. When this (rarely) happens, it's generally only small colonies, not adult corals. More typically, macroalgae abundance has been observed to increase as space is opened up on the reef after corals die.

We have deleted this sentence.

Minor Comments

Abstract:

P2, L5: change in to on

Updated

P2, L12: insert relative before dominance

Updated

P2, L17: comma after Here

Updated

P2, L23: change In to Over and add a comma after years

Updated

P2, L24-26: *Acropora* should be italicized and there should be a period after spp.
Updated

P2, L33-35: the final sentence is a fragment and should be combined with the previous sentence
Changed

Introduction:

P3, L15/16: I would add some references at the end of the sentence about the causes of coral decline

We have added the following:

“... The causes of coral cover decline include a combination of local and global anthropogenic impacts including overfishing, coastal development and associated pollution and rising sea temperatures [5–7].”

P3, L17/18: I suggest changing loss to decline and compromising should be changed to has compromised for parallel structure
Changed

P3, L19: It's not clear to me what the phrase “and hence diversity” means. Are the authors referring to the diversity of reef-associated biota? If so, please make that clear or simply remove this parenthetical

Changed to: L 48-51

“...This decline has resulted in the decline of the ecological performance of reefs and has compromised their future capacity to sustain their structural complexity (and with that the biota that depends of the structure), to maintain many ecosystem services and to keep up with sea-level rise [2,8–10].”

P3, L21 (and throughout): there should be a hyphen in sea-level rise
Updated

P3, L22/23; i.a. should be i.e. You could also cite Toth et al. 2018. Global Change Biology here.
Updated

P3, L28: I would break up the references, so it is clear which support the phrase about erosion/OA and which refer to hurricane impacts, since ref 7 was broken out in the first part of the sentence. You could also cite the newly published study by Kuffner et al. (2019, Limnology and Oceanography), which also discusses reef erosion on in the Florida Keys.

Updated L 52-55

“...These changes can occur either when vertical coral reef growth is halted or inhibited [i.e. reef 'turn off'; 11,12] when high rates of biological, chemical and physical processes drive net erosion of the underlying reef structure [13–15], or in response to direct impacts such as hurricanes through the breakage of coral skeletons[16].“

P3, L39: there should be a space between Pleistocene and the references
Updated

P3, L44: *Orbicella* is a single genus so, this should read: massive corals in the genus
Updated

P3, L45/46: “due” is missing after mainly
Updated

P3, L56/57: I would suggest citing at least one of Rich Aronson's studies (e.g., Aronson et al. 2004, Ecology) here because he was really the first to document increases in *Agaricia* populations following disturbance events. I would also make it clear in the next sentence that these taxa are relative dominants not the dominant benthos on the reefs in most cases.
Updated

P4, L13/14: missing the word “be” before reduced.
Updated

P4, L17: budget should be budgets
Updated

P4, L34/35: change between to in because here between suggests that surveys were done in years other than 1985 and 2016, which is not the case
Updated

Methods:

P5, L13: perhaps change associated with to determined by or driven by
Changed

P5, L14/15: change has to had
Changed

P5, L16/17: change is to was
Changed

P5, L18: change that to and
Changed

P5, L28/29: change in to on or along
Changed

P5, L30: comma before which
Changed

P5, L40: I'd change which to who
Changed

P5, L43/44: does “their distribution” mean the distribution of reef zones or the distribution of coral taxa across reef zones? Consider rephrasing.
Changed to
“... and generated detailed maps of the coral reefs distribution along the Mexican Caribbean coast.”

P5, L55/56: Change on to at. I'd also add a comma after five
Updated

P6, L17/18: I'd change this to the surveys

Updated

P6, L26-30: The wording of this sentence is somewhat confusing. I'd change to something like

We also include the reef crest in the back-reef zone because the transects surveyed in 2016 extend from the back reef to part of the reef crest. You may also want to reiterate that both zones are in similar depths.

The wording of the sentence was revised following your suggestions

P6, L33-35: It's not clear to me what the phrase "integrated into the reef zone level" means. Can you just say that transect data were summarized for each site?

The sentence was rephrased L 162-163:

"...The data at transect level was analysed by reef zone and used for the comparison between years."

P7, L37: either remove this was or make that the beginning of a separate sentence

Updated

P7, L60: I would consider making the name of this group more specific since other branching corals fall into different groups.

Since the genus *Acropora* has been important framework builders, following your suggestion we changed to "framework-building branching corals"

P8, L10: It would be useful to indicate at what size corals were considered non-framework builders instead of massives.

Our classification was based on a range of coral (not only in coral size or morphology), so although it is true that small corals tend to be classified as non-framework builders (or opportunistic or weedy), it is not possible to define a general threshold based only on sizes. Please note that our groups were proposed following Darling et al. (2012), Perry et al 2014 and Gonzalez-Barrios & Alvarez-Filip (2018).

P8, L55/56: there's an extra space between ^-2 and you're missing a space before year

Updated

P8, L57/58: I'd either add a reference to Gonzalez-Barrios & Alvarez-Filip here or add something like "as described below" in a parenthetical.

Updated

P9, L5: to should be from

Updated

P9, L7: by should be as

Updated

P9, L9: missing a space after skeleton

Updated

P9, L23-25: the wording of the final phrase of this sentence is awkward. I'd make it a separate sentence and say something like: The modified estimates of calcification rate calculated here represent...

Changed as suggested.

P9, L24-26: You've already stated the units for calcification rate twice in the previous paragraph: there is no need to repeat that here.
Updated

P9, L28: I'd change by to on each or for each
Changed

P9, L37: add (RFI) after Reef Function Index
Updated

P9, L39: I'd rephrase of each variable to for each of these metrics. Please also include a statement about how the data were standardized (by mean?)

Thank you for your comment, this was modified and we explain with more detailed the standardization: L 241-246.

"The mean estimates of calcification rate, rugosity, and size of each species were scaled using the minimum and maximum value of each variable as: $X = (x - \text{min value}) / (\text{max value} - \text{min value})$, where x is the value of each variable of each species. This standardization allows variables to have equal ranges (0–1). Then the three standardized variables were averaged to obtain a species-specific functional coefficient (Fc)."

P9, L42-46: I think it makes more sense to move this sentence to the end of the paragraph
Updated

P9, L60: need a d at the end of change
Updated

P10, L11/12: need a space after Caribbean
Updated

P11, L24: Change reef functional index to RFI or at least capitalize for consistency
Updated

P11, L27: change this to which or make a new sentence
Updated

Results

P11, L39-42: I'd add 95% before confidence intervals. Also, in some places Supplementary Table is fully capitalized and in other places table is lowercase.
Updated

P11, L48: Again, I would indicate that you're talking about **framework-building branching** corals here.
This was corrected following your previous comment and thorough the document

P11, L55: significant should be significantly
Updated

P11, L58/59: builder should be building
Updated

P12, L7: I'd change being to and was
Updated

P12, L52-54: the phrasing of this sentence is awkward. I would combine with the previous sentence: and between groups...you also need a comma before although.

Updated

“... A two way crossed ANOSIM, showed significant differences between sampling years across reef zone groups ($R= 0.624$, $p >0.05$), as well as differences between reef zones groups across years ($R=0.338$, $p = 0.05$).”

P12, L59: replace smaller with smallest

Updated

P13, L40: should read non-metric multi-dimensional scaling (nMDS)

Update

P13, L41/42: MBRS hasn't been defined anywhere in the manuscript

This was changed to “Mexican Caribbean”, to be consistent with the description site.

P14, L19-21: Why is the data presentation different here? You're reported means \pm CI elsewhere already.

Update

P14, L30/31: delete actual

Update

P14, L34: comma after In contrast

Update

P14, L41: I'd rephrase this to: ...1985, we found that in 2016...

Update

P14, L45-48: I'd add “on average in the back-reef zone” after negative. I would also delete “given that” from this sentence.

Update

There were several places in the Results and Discussion where it was not immediately clear to me which reef zone you were talking about

Thank you for your comment, it was reviewed and if necessary emphasize the correspondent reef-zone.

P14, L50: You're just talking about the back reef here too right? I would add that before transects.

This was re-written to emphasize correspondent reef-zone

P15, L35: add 95% before confidence intervals

Update

Discussion

P16, L9: add a comma after back-reef

Update

P16, L32: change in to on

Update

P16, L58: I'd change space to niche

Update

P17, L14-21: You need to provide context for where this study was conducted, otherwise I think discussion of COTs, which aren't found in the Caribbean, is confusing. This is also just a very long sentence, so I would consider breaking it up.

Thank you, since the COTs aren't found in the Caribbean, we removed this example.

P17, L33: I would rephrase this sentence to something like: ...back-reef at our sites in 1985 was already well below this (add G from your site here), but...

Modified

P17, L41/42: I would say production was fairly negative, not neutral in the back-reef environments by 2016.

Modified

P17, L51: perhaps change discussed to proposed

Updated

P17, L57: addition should be additional

Updated

P18, L21: hyphen in sea-level rise and erosion should be erosional

Updated

P18, L23-24: isn't loss of three-dimensional complexity the same thing as breakdown of reef structural complexity. Delete this phrase

Updated

P18, L33: comma before which and after area

Updated

P18, L45/46: need a space after conditions

Updated

P18, L52/53: you need a citation at the end of this sentence

Updated

P18, L59/60: has should not be italicized

Updated

P19, L3/4: need a space after reefs

Updated

P19, L19: add citations at the end of this sentence

Updated, we include "Perry 2014; Januchowski-Hartley 2017"

P19, L12: you may want to cite other studies that have found similar results here. See see Bruno et al. 2019. Climate change, coral loss, and the curious case of the parrotfish paradigm: why don't MPAs improve resilience? Annual Review of Marine Science and Cox et al. 2017. MEPS. Establishment of marine protected areas alone does not restore coral reef communities in Belize

We have removed this sentence

P19, L40/41: I'd move "in the future" to the end of the sentence

Update

P19, L42: modifying should be changed to being modified or simply changing
Update

P19, L45/46: on coral reefs is repetitive. I'd delete. I'd also put a comma after the citation
Update

Reviewer: 2

Comments to the Author

This manuscript describes changes in the coral assemblage on the Mexican reef Puerto Morelos since 1985 and the repercussions these changes have on gross and net carbonate production and reef structural complexity. The authors have looked at sites within the back-reef and fore-reef zone and especially in the former the coral community has changed considerably. Compared to 1985 branching (mostly *Acropora* spp.) and foliose coral species have strongly declined in cover and non-framework building species have become a proportionally important group among corals. The authors state that both the relative shift in groups within the coral community as well as the general decline in coral cover has strongly impacted the capacity of these reefs to maintain net carbonate production. Although I appreciate the work conducted by the authors and the challenge they faced comparing novel data to incomplete historical data (e.g. no information on bioerosion) I have a few concerns about the presented work.

The authors describe the change in the coral community based on major groups distinguished by various life-history traits. The authors only haphazardly describe some changes in specific coral species. Although it makes sense to look at functional groups for some analyses and comparisons I think a lot of information is lost by not looking at the individual coral species as well. In the branching group it is clear that the loss of Acroporid corals has resulted in a considerable decline, but now it is unclear how, for instance, *Orbicella* spp. have changed. On the back-reef there appears to be a small, albeit not significant, increase in cover of massive species. It would be very interesting to find out the massive species found in 2016 compare to 1985. I would maybe expect more (*pseudo*)*diploria* spp. The same goes for the non-framework building corals, do you find the same species in 1985 and 2016? This would also be relevant in light of the described homogenization, maybe this is true for groups but not at all for specific important species.

Thank you for this comment. We would like first clarifying that we conducted three approaches for this manuscript. First we used functional groups, as this approximation has been proven to be very useful in summarizing community dynamics that otherwise could be confusing (for example when using a long list of species; see for example McGill et al., 2006, Trends Ecol. Evol; Cadotte et al 2011 Journal of Applied Ecology). Also, a functional groups approach can be useful to quantify, predict, and better anticipate, the impacts of disturbances on ecological communities (e.g. Mouillot et al., 2013 Trends Ecol. Evol). Our second approach consisted of exploring changes in the community composition (Figure 3 and Table 1). This analysis reflects the changes in the composition of all coral species. The third approach aimed to describe the functional consequences of the observed changes in the previous two approaches.

We did not emphasize on individual species trends because we wanted to keep a clear and direct message, but we do highlight those species that were responsible for the most evident patterns in our findings. Particularly, changes in the cover of *Acropora palmata* were very important in the observed declines in the back-reef sites (for example, see lines 315—318). Declines of *Agaricia tenuifolia* and *Porites porites* (the other two species that showed clear trends of decline) are also adequately mentioned in the manuscript (lines 318-321). *Orbicella annularis* was not very abundant in our study sites in 1985 and therefore the contribution to this species to the observed trends was not as relevant as we would have expected. The non-significant increase observed in the 'massive' species is the result of slight increases of a few species (mainly

Orbicella annularis, *Pseudodiploria strigosa*, and *Siderastrea siderea*. We have now made this clear in the manuscript (lines 324-326).

Somewhat in line with this issue I think it is necessary to provide a more detailed description of how these reefs looked in 1985 and even before. The authors touch upon a number of relevant issues associated with reef degradation, including the loss of positive net growth, structural complexity and shifts within the coral community. Although all these changes indeed impact reef functioning throughout the Caribbean region they seem less relevant to the reefs described in this manuscript. If I understood correctly, net reef carbonate production was, on average, already negative in 1985. This would mean that since that time reefs have been losing structural complexity? I think it would be very useful to provide some information on the carbonate production and erosion factors. Was there for instance a lot of erosion in 1985? And if so, by which organisms? I am surprised to see there is such low carbonate production if there used to still be a considerable cover of *Acropora* species which are known to have fast rates of carbonate production.

Unfortunately bioerosion data for 1985 was not available and hence we assume that the bioerosion from 2016 was the same as in 1985, this assumption is supported by several observations listed on the response to comment 4 of reviewer 1. See the revised version of methods (L 251-277) and Supporting Figure S2

Also, it is now unclear if the authors had rugosity data for 1985. A short section describing the historical reef morphology, reef community, etc. of these reefs would help to place the current findings in a historical context.

We have expanded the description of our site (please see the study area section).

We did not use reef-scale rugosity in our calculation. To account for the structural complexity of coral species (and their morphology) we used the method proposed in González-Barrios & Alvarez Filip (2018), which provides estimates of species calcification rate considering each species' morphology and structural complexity. A detailed description of this approach is provided in the methodology section lines 235-250

The authors point out that they found considerable variation in carbonate production both in 1985 and 2016 among sites. This implies a lot of variation on the studied reef stretch. If this is the case, I am wondering if the authors surveyed enough reef sites to come to a representable mean. I realize that the number of sites is limited by the sites surveyed in 1985, but maybe some more justification is needed. I would also recommend to use boxplots in figure 4 and also show the original data points in it. This would give a much clearer view of the spread within the data. It would maybe be interesting to show the individual transects as point. Within the text it could be better to give ranges with the averages that the authors provide, especially for carbonate production, which can differ considerably between transects/sites.

The variation observed in our results is in part due to the high cover of *Acropora* in some transects. This has been included in the results section "... Although in 1985 net carbonate production was also negative on average in the back-reef zone, the variation between transects is very large due to the high cover of *Acropora* in some transects, while in other transects coral cover was already very low (Fig. 2)". It is important to note that this 'patchy' distribution it is inherent to the natural distribution of *Acropora palmata* stands in the back-reef/fore-reef environments. As suggested we have also included

the original data points (transects) in figure 4. This helped to make clear the effect of those transects with high cover of *Acropora*.

Following up on this the authors should consider including a supplementary table with values for carbonate production (maybe separated by functional group), estimated erosion by group of organisms, rugosity, etc. on a transect level.

Following yours and Referee # 1 observations, we have added to the Supplementary material a table with summary information of carbonate producer and bioeroders for each reef zone.

Overall, the manuscript often does not read very smooth. Especially in the introduction and discussion, it seems the authors jump from thought to thought where more explanation is needed (e.g. page 3 line 17).

Thank you for your comments, the wording and fluency of the text was revised.

Following your comments and those from the Referee # 1, we have reviewed the entire Discussion in order to provide a better structure, avoid redundancies and make a clear distinction between the findings of our study and findings described in previous literature.

I feel that especially the discussion could do with a better structure. The authors should make a clear distinction between the findings of this study and findings described in previous literature, right now this is not always very clear what their novel contribution is. Normally I don't think a general comment like this is very useful so I will try to give some examples:

"The increase of these non-framework species had no measurable effect on the functional potential of the fore-reefs as these species contribute very little to the reef structure and carbonate production [36]."

"This modification of coral communities has led to a biologically homogenization between reef zones, whereby instead of a dominance of reef-builder coral species (a situation which was both historically and geologically the norm in the Caribbean; [18,81,82]), there are more non-framework species that cannot fulfil the same functions as reef-builders, leaving an important space vacant."

Thank you for taking the time to highlight these examples. We have amended all of them.

In part I think this has to do with the build up of sentences. They are often very long and in multiple occasions the start of the sentence does not link properly to the end. Here are some examples of difficult sentences and some in which I feel the cause-effect is not fully correct in my eyes:

*"The resultant decline of the major reef-building coral species across the Caribbean has led to a relative increase in the abundance of non-framework coral species, such as *Agaricia* spp. and *Porites astreoides* [32–34]."* Is the decline in RB species indeed the main cause for the increase in non-framework building species?

No, the redaction was corrected as follows:

*"... The resultant decline of the major reef-building coral species across the Caribbean has been accompanied by a relative increase in the abundance of non-framework coral species, such as *Agaricia* spp. and *Porites astreoides* [37–40]"*

“The ultimate consequence of a reduced abundance of important reef-building species will be reduced reef-carbonate budgets, with rates of bioerosion becoming increasingly important controls on overall budget states” An ultimate consequence is not that bioerosion becomes increasingly important. Consider splitting the sentences.

The redaction was corrected as follows:

“... The ultimate consequence of a reduced abundance of important reef-building species will be reduced reef-carbonate budgets. Along with this the decline, the rates of bioerosion may become increasingly important controls on overall budgets states [44,51].”

“By considering the characteristics of each species (morphology and growth), potential overestimations of calcium carbonate production are avoided, and thus represent the contribution of habitat forming species to carbonate accumulation.”

Changed to: “... By considering the characteristics of each species (morphology and growth), potential overestimations of calcium carbonate production are avoided.”

As a general point I wonder what the effect of declined coral cover is on the issues described in this manuscript. The main focus now lies on the shift in coral communities. And although this indeed has a clear effect, but figure S1 shows an overall decline in cover of about 50% in the back reef. I think that this has a significant impact on for instance carbonate production as well. Yet this is not really covered in detail in the discussion.

We have included mention to overall coral changes in Results (Lines 310-318) and Discussion (Lines 379-383). But overall, the main message is that the observed decline in the coral cover is mainly due to the loss of framework-building branching corals, which lead to a significant impact on the carbonate production.

Minor comments by line. I apologize for any inconsistencies regarding line numbering. In the document I received line numbers did often not align with the actual sentence.

Page 2

Line 15: ‘function’ may on itself be a bit vague.

Corrected, we refer to the physical reef function.

We also include an explanation on what we mean as the “physical reef function” in the introduction lines 100-101.

Line 18: shortly mention how the coral communities were evaluated (method)

The following was added: “..We used the cover of coral species to explore changes in four morpho-functional groups, coral community composition, coral community calcification, the reef functional index and the reef carbonate budget.”

Line 24: *Acropora* in italic (see more occasions throughout the manuscript)

Updated

Line 33: it feels like the last sentence should be linked to the previous or a verb needs to be added

This was re-written.

Page 3

Line 18: compromising should be compromises

Updated

Line 27: do hurricanes not also cause physical erosion?

Yes this idea is now included in the discussion. See the response to the first comment of Referee #1, and lines 470-475

Line 47: Mainly disease? Bleaching through thermal stress also had a major effect I assume

Indeed, the text was modify to include bleaching.

Page 4

Line 14: reduced (and space after).

Updated

Line 16: example of a sentence that could be more to the point: for instance: *will transition into states of net erosion*

Updated

Line 21: Maybe personal, but *reef structure* could mean a lot of thing, structural complexity?

Changed

Line 41: are the groups really only distinguished by colony shape?

No, the function provide to the reef-framework building was also taken into account. See L 173-186:

“...We distinguished four main groups of corals based on colony morphology and their contribution to reef framework [44–46]: (a) framework-building branching corals - specifically the historically important reef framework building *Acropora* species; (b) massive species form the second important group of reef-framework species, in this group *Orbicella* is the main reef-builder genus, but we decided to included it along the other massive species because their contribution to the overall coral cover is relatively low; (c) small non-framework builder species, which some authors define as opportunistic, are small species that do not contribute greatly to calcification nor the structural complexity [44,45], and (d) foliose-digitiform species (*Agaricia tenuifolia* and *Porites porites*), these species are considered as part of the opportunistic group by some authors [44,46], but we decided to treat them as separate groups because of their contribution to reef three-dimensional structure at fine-scale creating important microhabitats and are susceptible to breakage (thus generating rubble), also this group is highly represented in this zone [43,65].”

Line 45: What exactly do the authors mean with physical reef functioning

The capacity of the reefs to sustain future reef-framework, carbonate production and reef accretion. This was clarified in the introduction. L 100-101

Page 5

Line 8: consider including a lead sentence introducing the study site Puerto Morelos, now it comes a bit out of nowhere

Corrected

Line 10: Is it both a fringing reef and a barrier reef?

No, these reef are referred as extended fringing reefs, as these reefs do not form a classical barrier reef Jordán-Dahlgren & Rodríguez-Martínez (2003). In the text this was modify.

Line 15: what do the authors mean here with “mostly associated”

Changed to “...mostly driven by wave exposure and light penetration”.

Line 16: “Historically it has a well-developed back-reef and reef-crest that were dominated by *Acropora palmata*” What about *cervicornis*? I am just interested to know. Also, what does this mean for carbonate production? It seems that a reef dominated by *palmata* should have a considerable positive net production. I think here you could use the known literature to describe a bit more the reefs at Puerto Morelos, also historically. The study area section was complemented with a more detailed description of the Puerto Morelos reefs.

As for the carbonate budget, since we are using bioerosion rates of 2016, this could be underestimated (see the response to comment 4 of reviewer 1).

Line 28: ‘in the coast’  in the coastal area?

This was changed following the comment of Reviewer 1, for “ along the coast”

Line 30: this sentence is a bit long and becomes confusing, also could the authors provide some more detail on the sewage status? Where does this go? Does it really all seep in? what about run-off? Untreated discharge through pipes?

The paragraph was reworded.

Line 56: here the max depth is 20 m, but before the authors talk about 30 m

This was only for a few sites from the south of the Mexican Caribbean; in order to not confuse the reader we homogenized it to 20-25 m depth.

Line 57: It is not entirely clear how the transect where placed. Where 20m transects placed in each zone? Or did a transect cover multiple zones since they were placed perpendicular?

The transects did not cover multiple zones since the objective was to assess each reef zone individually, we corrected the redaction as follows: L 147-148 “...At each zone (back-reef and reef-crest) or depth in the case of the fore-reef, five, 20 m long transects were placed haphazardly, perpendicular to the coast, separated from each other by 5-25 m.”

Also, how haphazardly were transects placed in both years? How was the start point determined?

The start point was determined with the beginning of the reef structure, and the transects were placed following the reef structure.

Page 6

Line 9: the applied methods in 1985 and 2016 are very different. If there is enough data I think it is ok to compare, but the authors should be very careful with drawing strong conclusions especially because it seems there is considerable variation within the reef. Maybe it would be good to also mention the potential shortcomings of this in the discussion?

Previous studies have shown that these two methodologies are fairly comparable. See Beenaerts & Vanden Berghe, 2005 West Indian Ocean J Mar Sci; Facon et al., 2016 Ecological Indicators; Darling et al., 2017 Coral Reef). Please also, see our response to comment 2 of Referee # 1.

Line 17: I understand they may not have collected GPS data, but it was not pre-GPS. Formulate a bit different, maybe: ... before the general use of GPS in scientific research...

Update

Line 25: why only these two zones?

We choose to analyse the reef zones to explore how did the coral community changed between reef zones, in the end we only have the back-reef and fore-reef zone, but it is

important to say that the back-reef and reef crest were considered as one zone, due to the methodology of 2016, and because these zones have similar species composition, habitat and depth in the studied sites. L 157-160

Line 35, remove comma before and.
Updated

Figure 1

- Font of coordinates is too small
Corrected

- Why is site 1 located so far from sites in 1985?
Since we used historical data we tried to match the sites according to their characteristics (depth, species composition). Another point to consider is that the rectangles corresponding to sites in 1985 are an approximation, since the geographical coordinates were not available. But despite the distance we consider that there were similar conditions in both sites

- I don't think symbology is the correct word
Corrected

Page 7

Line 52: it is not clear why the authors go from classifying species to persistent through time.

This sentence has been reworded.

Line 55: this part I miss a bit in the results, the specific changes in specific coral species

Here we aim to explore changes in coral community composition, not specific trends by species. Following one comment of Referee # 1, we have broadened the description of this approach in the methods section (Lines 188-207).

Page 8

Line 3: Only acropora? What about *Orbicella*?

Orbicella is also an important reef-framework building species, but since *Acropora* played an important role on the physical structure in the back reef zone, we decided to make a separate group with only the genus *Acropora*.

Line 35: be careful throughout the manuscript not to use too many unnecessary abbreviations and if abbreviations are used be consistent.

This was reviewed throughout the document.

Page 9

Line 5: space between units
Updated

Line 26: estimated **and expressed** in units?
Corrected

Line 32: I partly agree, but the calcification by CCA should not be underestimated, especially in the cryptic environment and in 1985. If the cover was measured you could consider including it or mention that it was actually low in both years but then provide a value.

The cover of CCA was not measured for 1985, and in 2016 the contribution from CCA to the carbonate Budget is really low we decided to remove this.

Line 37: is there rugosity data for 1985?

No, the rugosity data was taken from Gonzalez-Barrios & Alvarez-Filip (2018), and it was at the species level. This was also applied for 1985 data for the reef functional index.

Line 37: example of abbreviation (this is the first mention of RFI right?)

Corrected

Line 39: how were the data standardized?

This was included in the description: "... The mean estimates of calcification rate, rugosity, and size of each species were scaled using the minimum and maximum value of each variable as: $X = (x - \text{min value}) / (\text{max value} - \text{min value})$, where x is the value of each variable of each species. This standardization allows variables to have equal ranges (0–1). Then the three standardized variables were averaged to obtain a species-specific functional coefficient (Fc)."

Line 42: the following section needs some more elaboration, for the reader it is not fully clear what applies to the RFI or the Fc.

Thank you for your comment. The description was expanded.

Line 47: not clear what the authors mean here with 4th root. Was the data transformed? I was wondering about this in general. Did the authors consider transforming the data?

The 4th root transformation is made on the Index, to rank the index from 0 to 1, to have a better interpretability of the data (Gonzalez-Barrios & Alvarez-Filip, 2018).

Page 10

Line 12: Space between Caribbean and refs

Updated

Line 24: abundant that  abundant than

Updated

Line 35: I realize it is impossible to retrieve data on bioerosion, but this section includes a lot of assumptions. What about excavating sponges? They can be important eroders, especially on Caribbean reefs and their abundance increased in recent years.

We agree in that sponges are important bioeroders. Following your comments and some suggestion of Referee # 1, we now mention their importance and the fact their effects of bioerosion may be increasing in the methods (253-277) and the discussion (461-464). Please also note that we now provide a new supplementary table (Table S5) with the observed bioerosion rates for this group (and others).

Line 52: The value for sponge erosion is outdated. The reef budget method at present also includes species specific rates for sponge erosion. Otherwise also see De Bakker et al. 2018, PlosOne.

We are in the process of updating the ReefBudget methodology for the Caribbean with all this newly available information. However, this is not yet finished. Besides, we preferred to use the current methodology (and rates) as our results could be directly comparable to all other studies that have used this approach in the past.

Page 11

Line 10: I thought more species specific rates were available

Same as previous comment.

Line 14: which constants were used? 0-5 or 5-10 for the different zones? Maybe also point out the approximate depth range of the two studied zones earlier.

As pointed in the methodology section, for the back reef the depth range between 2-5, so we used constants 0-5, and for the fore-reef the depth range between 6-13 so we took into account the 5-10 m constant. The depth range is in the data collection section L 152-154.

In the coral community section it would be very interesting to include some species specific information.

What reef builders do now dominate the back-reef? Are they the same as in 1985? More *Diploria*?

The changes in the community composition of the back-reef were largely defined by the drastic decline of *A. palmata*, but also by the declines observed in *P. porites* and *A. tenuifolia* (see lines 315-321). The cover of the rest of the species changed very little, and currently, a mixture of massive and weedy species delineates this zone. Please also see our answer for your first comment.

Page 12

Line 52/52: a general remark, consider presenting statistical characters in italic and be consistent with space in presented p values (e.g. $p = 0.01$). See for instance also Page 14 line 34, here the authors use = and <.

Corrected through the text

Figure 3:

- No purple or green colour, I only see grey and black
- write out MBRS

Updated

Page 13

Line 13: make sure it is clear what kind of homogenization you are talking about, among sites, within sites, among zones. It will not always be correct to talk about homogenization since it seems that there is a lot of variation still between sites and the actual contribution of the various groups compared to the relative contribution.

Thank you for this comment. We based this conclusion largely on the coral community composition analyses (Fig 3, Table 1) which clearly shows a higher similarity between reef zones in 2016.

Page 14

In the section starting at line 40 it is very hard to understand what the authors want to say. Also it seems that the sentences are not correctly written.

The wording of the paragraph was reworded.

Line 45: here the authors should mention why it was already negative and that there was a lot of variation in 1985. This is a strange finding without explanation and could question the relevance of this study. If the reef was already disappearing then what has changed?

Thank you; please refer to our response to your comment # 4.

Line 51: still significant yes but the authors don't claim it changed significantly, so why the 'but....'

The word but was misused and deleted.

Figure 4:

- Consider the use of boxplot with the actual data points
 - explain what the authors mean with 'height'
 - I assume these are **95% CI**
 - explain what the stars mean (I assume significant change?) Also in figure 2
- Corrected. The plot now contains original data following one of your first suggestions.

Page 16

Line 15: This seems somewhat strange, how was this effect measured? I am sure that a decline in rugosity while cover remains the same (with different species) could be measured. If not, I wonder if the authors can make this claim based on their data.

One of the aims of this work was to assess the changes in the reef function that could have happened as consequence of the change in coral species composition. For this three variables were taken into account: the coral community carbonate production, the reef functional index and the net carbonate budget. Based on these measures we conclude that the increase of non-framework species did not affect this functional potential, because they contribute little to the carbonate production and the future reef-structure (due to the size and growth they have).

Line 24: is there evidence in the data that this will happen?

Our results show that in 2016 carbonate budgets are in negative state, meaning that there is no accretion that could maintain the reef-structure if bioerosion continues to increase, so we infer that if current conditions do not change, this could happen. Also other authors have found similar results with the increase of non-framework species (Perry et al., 2014, *Global Change Biology*; De Bakker et al., 2016 *Frontiers in Marine Science*).

Line 25: they were already negative in 1985. This sentence implies they are currently in a negative state but were not before. Also I am not sure if it is correct to make a claim on a change in the impact of bioerosion because bioerosion was assumed to be similar in both years.

The paragraph was reworded.

Page 17

Line 5: *Along with the loss of massive-framework species, climatic factors are changing the coral community assemblages.* Is the loss of massive species not also the result of climate change and local impact?

Indeed, the loss of massive species is consequence of local impacts and climate change. We have reworded this to emphasize that along with the loss of some species, other species are responding differently by expanding their ranges.

Line 9: I wonder if the move northwards is really a relevant issue here, north of the studied reefs the *Acropora*'s have seen massive mortality as well.

We consider it's pertinent in terms of allowing the reader to analyse that the change between reef zones isn't the only consequence of climatic factors, but also the expansion of habitat range.

Line 14: does this apply to the Florida reef or the data collected for this study? And here it is not clear what kind of homogenization is meant. Are we now talking about homogenization on a species level?

This has been re-written.

Line 20: the crown-of-thorns-starfish is not an issue in the Caribbean right?

No, this example was removed.

Line 23: is the spatial variation based on species composition or on the groups.

Reworded.

Line 23: I don't see why spatial variation on itself is responsible for the potential loss of carbonate production

This has been re-written to clarify the idea as follows:

"The net loss of potential to accumulate CaCO₃ reported here compromises the ability of coral reefs to sustain high rates of reef accretion, especially in the back-reef, which was previously the best developed zone in the north section of the Mexican Caribbean due to the contribution of the genus *Acropora* [55,98]."

Line 27: elaborate on the meaning of well developed.

See previous comment

Line 37: I don't think this is what it suggests.

Changed to "...This suggests that the reefs had already shifted towards net negative (and thus potentially net erosional) states before the start of our study period – a transition also suggested in recent work from Florida [10], although this needs to be taken conservatively in our case as we assumed that bioerosion rates in 1985 and 2016 were similar (see methods)."

Line 42: What do the authors mean by neutral?

Changed to "fairly negative"

Line 43: fore-reefs

Updated

Line 45: How historically? Is there data from before 1985. Has there never been a well developed reef there?

We have improved the description of the study area.

Line 57: additional issue

Updated

Page 18

Line 3: Yes, but what about *Orbicella* spp. in your data?

Our data show little change in the cover of massive species in general and *Orbicella* in particular.

Line 11: Yes, but is this an issue of your reefs if they have never had net positive carbonate production

The point here is the great variability in 1985 (that now is clearly described because of your comments). The mean carbonate budgets in 1985 were non-significantly different from zero, which suggest that production and erosion were in balance. By 2016 the budget is clearly negative.

Line 12: yes, the rubble is a point, but is it relevant to your study? Now it comes a bit out of nowhere.

The export of rubble and sediments from one reef to another is important in the future formation of new structure (Blanchon et al., 2017, e.g. *Frontiers in Marine Sciences*).

Line 15-19: True, but again this seems not too relevant for the reefs described in this study because they were already negative.

Please see our answer to your comment in Line 11.

Line 30: I agree these kind of reefs can serve as a buffer, but maybe the authors should elaborate a bit on the reasons why. E.g. recruitment, etc.

Following your and reviewers 1 suggestions we enhanced this part of the discussion.

Line 47: The issue that the community will not go back is more related to the idea that the impact on reefs nowadays will not likely become better, rather it will likely become worse.

This has been removed

Line 49: what do the authors mean with simplified communities? A community dominated by *Acropora* corals is in my opinion more simple than a community with many different organisms.

Simplified communities in terms of the contribution they made to the reef function. We rewrote this part.

Page 19

Line 19: Is nutrient enrichment not also caused by large by the absence of functional waste water treatment? Other stressors are more temperature, sedimentation, etc....

This line was removed

Line 24: *overwhelming the available space that will eventually overgrow corals*. This sentence needs to be changed now it seems the available space is overgrowing corals.

This line was removed

Line 30: **composition of** the coral assemblage

Updated

Line 39: and vice versa, other organisms can also cause decline or recovery of reef-builders

This line was removed

I would like to point out that I believe this work describing temporal changes in reef carbonate production potential can be a valuable contribution when aforementioned concerns are addressed in an appropriate way.

Reviewer: 3

Comments to the Author

Supplementary Information.

Table S1.

The CORAL TYPE COLUMN has a mixture of morphological information and function (nonframework and presumable framework). For clarity you can separate information on function and coral morphology type. Some of the species designated as Non-framework are not framework at their sites, but they are part of the framework at other Caribbean locations. (*Madracis*, *Agaricia*) are an integral part of the framework of many Caribbean reefs.

Thank you for this comment. Please note that we have now expanded the definition of each of the functional group in the methods (Lines 173-187). Also as suggested, we now include a new column in table s1 with the morphology of each coral species.

Also, we would like to mention that, for corals, morphological attributes are linked to their capacity to sustain reef framework, a positive carbonate budget and the potential of reef accretion (what we defined as 'physical functionality' in the manuscript; see lines 100-101). Coral-reef researchers have been increasingly using this approach and our classification corresponds with those used by previous studies. A baseline for our classification is Darling et al. (2012 Ecology Letters) but we adjusted the categories accordingly to Perry et al 2014, and González-Barrios & Alvarez-Filip, 2018. The species of the foliose-digitiform group (*Agaricia tenuifolia* and *Porites porites*) are considered as weedy/opportunistic by some authors, but we decided to include them in a separate group because they provide three-dimensional microstructure at lower scales, and as you pointed their contribution to the production of calcium carbonate is relatively important in some reefs (although not comparable with *Orbicella* or *Acropora*; see for example Aronson et al., 2004, Ecology; Toth et al 2019 Ecology).

Pg 3 lines 23- 27 *The ecological homogenization of the reef is provoking observation. The change is largely the consequence of the mortality of Acropora spp. so the reef has changed from having mostly large branching corals and foliose digitiform coral species in the back-reef zone.*

Thank you for this comment; we based this conclusion largely on the coral community composition analyses (Fig 3, Table 1) which clearly shows a higher similarity between reef zones in 2016. But we agree that the decline of *Acropora* largely explains these results. This is now clearly stated in lines 315-318.

26 add which foliose-digitiform coral species you are referring to in (parentheses).

Updated

28 Non-framework species. *Agaricia* spp. could in fact be a framework species. Consider including all of the *Agaricids* as foliose then recalculate the changes in accretion and homogenization.

Thank you for the observation and please refer to our response to your first comment. We acknowledge that some species that regularly are considered non-framework builders are indeed important carbonate producers (and provide fine-scale complexity) and this is why we decided to keep *A. tenuifolia* and *Porites porites* as a separate group in our study. In addition we would like to mention that although the other species of the genus *Agaricia* (and other species that we considered non-framework), could have high calcification rates, they have not been identified as major contributors of the reef-framework by ecological and geological studies across the Western Atlantic (see

for example: (Jackson, 2002; Budd & Johnson 1999; Aronson & Precht 2001; Precht & Miller, 2007; Pandolfi & Jackson 2006; Aronson et al., 2004; Jackson et al 2014).

30 -32 Calcification rates are decreasing. You explain nicely. Certain species that are decreasing were faster growing. Is there an overall measured decrease in calcification of species that are persisting? Also, the % of coral cover has decline and therefore calcification and accretion (eg loss of 3D structure).

In table S5 we now present the carbonate production by functional groups in which the changes in coral calcification can be observed. As described in the Results the main changes in the coral community calcification are explained by the decline of *Acropora* (and in minor manner *A. tenuifolia* and *P. porites*; lines 315-321). The coral cover decline for the back-reef was mostly to the loss of the genus *Acropora*, which has a major repercussion on the carbonate budget (lines 365-367).

PG 8 Line 58- Clearly describes all of this is this morphological classification should be included in Table S1 for consistency.

Table S1 was improved to complement the information.

P 10 Lines 7-27 Intraspecific calcification can be highly variable. Calcification is also a function of water depth and habitat (energy levels- eg leeward vs windward sides of islands).

Agree and have added the following in the methods L 226-230.

"...Rates of coral calcification are also dependant of local environmental conditions (such as light, depth or temperature; [71]); we however did not account for this source of variability into our analyses as local-scale information (e.g. skeletal density, growth rates) are not available for most of the coral species in our study site.

P19 Line 58 "addition" do you mean "additional"

Update

Appendix E

UNIVERSIDAD NACIONAL AUTÓNOMA DE MÉXICO INSTITUTO DE CIENCIAS DEL MAR Y LIMNOLOGÍA UNIDAD ACADÉMICA DE SISTEMAS ARRECIFALES

ADDRESS: PROL. AV. NIÑOS HEROES S/N, PUERTO. MORELOS, QR 77580 MEXICO
POSTAL ADDRESS: APARTADO POSTAL 1152, CANCÚN, QR 77500 MEXICO
TEL: 998 871 0219, 998 871 0009

12 September 2019

Dear Professor Jeremy Sanders,
Editor in chief, *Royal Society Open Science*.

We thank you and the reviewers for your comments on the manuscript '**Functional consequences of the long-term decline of reef-building corals in the Caribbean**'. The comments were very helpful and gave us the opportunity to improve our manuscript.

Below we provide details of how we have amended the manuscript in response to the reviewers' comments (shown in blue). We look forward to hearing from you and hope you will consider that following these revisions our manuscript is now suitable for *Royal Society Open Science*.

Yours faithfully,

Nuria Alejandrina Estrada Saldívar
(nuria.estradasaldivar@hotmail.com)

Dr Lorenzo Alvarez-Filip

Biodiversity and reef conservation lab (UNAM)
www.barcolab.org

Reviewer: 1

Comments to the Author(s)

Dear editor, dear authors

This is the second time I review this manuscript and I am very pleased with all the changes I don't see any major flaws in the updated version and the authors did a very good job addressing and changing all the points I raised before. I am happy to suggest publication of this manuscript after the final minor points as elaborated on below are addressed.

Minor comments:

Line 35: Point after *Acropora* spp.

Updated

Line 48: What exactly do you mean with "ecological performance", consider elaborating a bit more on what specific aspects of reef ecology are lost

We have deleted this part as it was redundant with the ideas we presented in the section of text.

Line 52: Space in front of i.e.

Updated as suggested

Line 58: With few I assume you mean a few species, not that there were not many framework building corals? It is not clear as it is currently written

Corrected as suggested

Line 69: consider removing "resultant"

Updated as suggested

Line 77: I would be careful with such a statement. Predicting what will happen within reef communities is near impossible. I would use words like "...we can hypothesize about future reef assemblages..."

Changed as suggested.

Line 79: is this indeed the ultimate consequence? Consider re-wording

Changed as suggested.

Line 113: "and descends gradually to a depth of..." this last section seems to not flow from the first part. Consider splitting.

Thank you. We have reworded this sentence.

Line 126: what about (artificial) beaches in this region? Are there any?

Not in our study area and therefore we did not consider it relevant to add any information regarding artificial beaches.

Line 146: Counted? Or was the cover under the line measured?

Changed as suggested.

Line 151: consider removing 'percentage'

Removed

Line 163: were analysed

Updated

Line 178: "...but we decided to included it with? the...."

Updated

Line 296: spp. After Echinometra

Updated

Line 300: Does the reefbudget method not include more parrotfish species

Yes, the RB method considers all the parrotfish species. We have reworted these lines to make this clear.

Line 325: might this have anything to do with the placement of the transects? How was the placement in 2016 chosen? Randomly? If not it could be possible that transect lines were placed on more developed reef sections, while some parts of the reef that used to be dominated by massive species may now have been transformed to sand or rubble patches.

The transects in both 1985 and 2016 were placed haphazardly within each reef zone. Please see lines 141-144 and 146-150. Also please consider that for the 1985 data the exact geographical location (coordinates) of sites were not recorded, but we tried to ensure realistic site comparison using only sites for which geographical location could be accurately constrained based on original maps and site descriptions (lines 153-157).

Line 340: be consistent with the spaces in your representation of $R= 0.6$ and $R=0.3$

Updated for consistency

Line 365: consider changing towe found, as expected,

Changed as suggested

Line 370: it would be useful to present observed ranges of net carbonate production next to the average production of the different zones.

The original data points (and therefore the range of variation) can be observed in figure 4.

Line 402: Not only a recent outbreak also previous ones and bleaching as you mentioned already in the introduction.

Agree, we have reworted this line.

Line 411: Space = niche?

Changed to niche

Line 431: consider replacing the – sign by a comma.

Updated as suggested

Line 434: No point behind spp and astreoides not in italic.

Updated as suggested

Line 435:have a limited....

Updated as suggested

Line 437: recognized should be recognize

Updated as suggested

Line 454: use either – signs or comma's to break this sentence, not both as in this case.

Updated as suggested

Line 459: fore-reefs sites, remove the s of reefs

Updated as suggested

Line 467: should be enhance

Updated as suggested

Line 470: space between building and [

Updated as suggested

Line 587: our recent publication (Extreme spatial heterogeneity in carbonate accretion potential on a Caribbean fringing reef linked to local human disturbance gradients accepted in GCB and in collaboration with Chris Perry) might be of interest regarding local variation in carbonate budgets and reef oases.

Thank you for your suggestion, we decided to include it as it is relevant for this study.

Reviewer: 2

Comments to the Author(s)

The clarity and organization of the manuscript is much improved in the revision and it is obvious that the authors carefully considered the suggestions of myself and the other reviewers. In particular, I appreciate the additional information about historic and present-day bioerosion in the study area, which justified their choice to use 2016 bioerosion rates for both modern and historical carbonate budgets. The flow of the Discussion has also been improved substantially and in general, the language of the manuscript is much more clear throughout the manuscript. There are a few places where the text could still use some rephrasing, which I have outlined in the Minor points below. Overall, I suggest that the manuscript be accepted to ROCS after some additional minor revisions.

The only significant concern I have, which I overlooked in the first review of the manuscript relates to the data analysis described in Lines 305-309. I'm wondering why the data weren't analyzed with a two-way test (i.e., a Kruskal Wallis test or an ANOVA with the data transformed or ranked) analogous to the two-way ANOSIM that was used to analyze the multivariate data? The major results of these analyses are likely robust, but testing separately for the effects of time and zones is not appropriate because these variables are not independent.

We agree with this comment and have corrected the wording of these sentences as we were only interested in testing differences between years for each reef zone (and not between zones). Please note that we do not present an analysis or a comparison between reef zones for these metrics.

Minor suggestions

Line 22: I'd Changed to "the last several decades" or just "recent decades"

Changed as suggested

Line 23-24: Add a comma after species and add "and" before leading

Updated as suggested

Line 34: Add period after spp.

Updated as suggested

Line 36: It might be good to add examples of the non-framework species whose abundance has increased as well.

Added as suggested

Line 41: sustain should be “sustains”

Corrected as suggested

Line 71: In the abstract you include *A. tenuifolia* as an example of a framework-building species. I would either Changed *Agaricia* spp. here to be more specific about what species are non-framework-builders (*A. agaricites*?) or just take it out and leave the *P. astreoides* example.

Updated to *Agaricia agaricites* and *Porites astreoides*

Line 99: Add the closed parentheses after “non-framework”

Updated as suggested

Line 110: add a comma after Historically

Updated as suggested

Line 113: add a comma after development and Changed “descends” to “descended”

Updated as suggested

Line 128: Changed “in average” to “on average”

Changed to “on average”

Line 132: add a comma after ref. 56

Updated as suggested

Line 134: Changed “coral reefs distribution” to either “coral-reef distribution” or “coral reefs’ distribution”

Changed as suggested to “coral-reef distribution”

Line 149: add a comma after zone and after eight

Updated as suggested

Line 151: Changed “at” to “to”

Updated as suggested

Lines 170-173: It’s not clear to me what the difference between the first and second analysis is based on this sentence. Is the first sentence referring to the functional group analysis? If so, make that clear.

Yes, the first analysis refers to morpho-functional groups and we have reworded the sentence to make this point clearer.

Line 177-180: I would simplify this sentence to something like: massive framework-building species, primarily *Orbicella* spp. as other massive taxa had low cover at our sites.

Thank you for this comment. This description was added following the suggestion of the first referee and therefore we would like to keep this level of detail. We however have reworded this sentence to make clear that the contribution of *Orbicella* to the overall coral cover is relatively low.

Line 181: I would add “which” before “are” to keep parallel structure
Corrected as suggested

Lines 183-184: Similarly, I would Changed “these species” to “which” and start a new sentence with “we decided”
Changed as suggested

Line 196: Perhaps Changed “width” to something like ecological space?
Changed to ecological space

Line 229: Changed “of” to “on”
Updated as suggested

Line 230: need a closed parenthesis after reference 70
Updated as suggested

Line 247: Changed “of” to “for”
Updated as suggested

Line 252: Changed “the” to “a”
Updated as suggested

Line 319: Acropora needs to be italicized
Updated as suggested

Lines 321-324: is this also in the back-reef zone? Please make this clear.
This results refer to the back reef zone, we reword it to make it clearer.

Line 323: “The two species...” should be a separate sentence
The sentence was separate.

Line 324: I’d suggest changing “Contrary” to something like “In contrast”
Changed to “In contrast”.

Line 325: “significant” should be “significantly”
Updated as suggested

Line 333: I would also add a list of what species actually increased in abundance in parentheses (P. astreoides and A. agaricites?)
Thank you for your suggestion, we include the species that increased.

Line 343: add a comma after (SEAc)
Updated as suggested

Line 350: I would consider changing to something like: was smaller in both reef zones in 2016 compared with 1985, as the coral communities became more similar across reef zones. Because “getting smaller” implies that the Changed is continuous and ongoing, which is not clear from your study.
Changed as suggested.

Like 367: Changed “provided” to “provides”
Updated as suggested

Line 374-375: This is not a complete sentence as currently written. Rephrase to something like: In contrast, only two transects of the back-reef zone has a positive, near neutral carbonate budget in 2016.

We reworded as suggested.

Line 377: Changed “which was” to “and the Changed between years”.

Updated as suggested

Also, are the carbonate budgets in the fore-reef significantly negative or does the uncertainty overlap with zero. Please make this clear.

The carbonate budget in both years was significantly negative. We have now make this clear.

Line 409: I'd suggest changing “builder” to “building”

Updated as suggested

Line 419: Changed the Burman et al. reference to its reference number

Updated as suggested

Line 422: Changed “cases” to “case”

Updated as suggested

Line 425: You're missing a space in Fig.4

Updated as suggested

Line 428: Add “have” before “led”

Updated as suggested

Line 433: You've used *Agaricia* rather than *Undaria* elsewhere. I think the community is back to using *Agaricia* now, right?

Indeed, we Changed it to *Agaricia*

Line 434: *astreoides* should be italicized

Updated as suggested

Line 437: Changed “recognized” to “recognize”

Updated as suggested

Line 452: I'd say “reefs in the Mexican Caribbean had...” for clarity

Updated as suggested

Line 456: I'd remove “In contrast” from the beginning of this sentence since the next sentence starts the same way

Updated as suggested

Line 479: Something is wrong with the wording here. Perhaps “since they are prone to mechanical breakage, which can lead to a decline in their cover.”

Changed as suggested

Lines 480-482: The language of this sentence is also not clear. May “On the other hand, low intensity storms can also have positive impacts on reefs, by cooling...”

Changed as suggested

Line 487: Changed “consider” to “considered”

Updated as suggested

Line 496: Is “actual” the right word here? Natural, maybe?

Changed to “current”

Also please Changed “Improve” to “Improving”

Changed as suggested

Figures 2 and 4 captions: In the last sentence “de” should be “the”

Updated as suggested

Figure 4: in the caption and in the text you say carbonate production, but the axis label is Coral calcification rates, which really isn't accurate. In the caption for b) I'd suggest saying “, which considers...” for parallel structure.

Updated as suggested

Table 1: what does the 2 superscript indicate?

The superscript was added following previous studies that have used this approach, as the idea is to indicate that the units refer to two dimension (an area). However, we decided to remove the superscript as this information is redundant with the description of the metrics.